# Reinforcement Learning Meets Masked Generative Models: Mask-GRPO for Text-to-Image Generation

**Yifu Luo** [†,‡,1]**, Xinhao Hu** [†,1]**, Keyu Fan** [†,1]**, Haoyuan Sun**[†,1]**, Zeyu Chen**[1,2]**, Bo Xia**[1]**,
Tiantian Zhang**[*,1]**, Yongzhe Chang**[1]**, Xueqian Wang**[*,1]

[†]Equal Contribution

[‡]Project Lead, [*]Corresponding Authors,

[1]Tsinghua University   [2]Technical University of Munich

## Abstract

Reinforcement learning (RL) has garnered increasing attention in text-to-image (T2I) generation. However, most existing RL approaches are tailored to either diffusion models or autoregressive models, overlooking an important alternative: masked generative models. In this work, we propose Mask-GRPO, the first method to incorporate Group Relative Policy Optimization (GRPO)-based RL into this overlooked paradigm. Our core insight is to redefine the transition probability, which is different from current approaches, and formulate the unmasking process as a multi-step decision-making problem. To further enhance our method, we explore several useful strategies, including removing the Kullback–Leibler constraint, applying the reduction strategy, and filtering out low-quality samples. Using Mask-GRPO, we improve a base model, Show-o, with substantial improvements on standard T2I benchmarks and preference alignment, outperforming existing state-of-the-art approaches. The code is available on `https://github.com/xingzhejun/Mask-GRPO`.

## 1 Introduction

Generative text-to-image (T2I) models [1–5] have made tremendous progress in recent years, offering powerful and innovative methods for visual content creation. Broadly, existing T2I models can be categorized into three main types: diffusion models [1, 6], autoregressive (AR) models [7, 8], and masked generative models (MGMs) [9–13]. Diffusion models generate images by gradually refining random noise through a denoising process, while standard AR models treat image generation as a sequential token-by-token prediction. MGMs, on the other hand, can be seen as a hybrid [10, 14]: they predict all masked tokens simultaneously in parallel at each iteration, but only keep the most confident ones, defined as newly unmasked tokens. The rest will be remasked for the next iteration, which we define as newly remasked tokens. This approach can be thought of as a discrete diffusion process through the absorbing state ([MASK]), progressively unmasking tokens while preserving the autoregressive nature, predicting masked regions based on predicted ones, similar to Bert [15]. This hybrid makes MGMs show superior trade-offs between sampling quality and speed [13].

Meanwhile, reinforcement learning (RL) [16–18] has gained increasing attention for its strong capability to enhance the reasoning capabilities of Large Language Models (LLMs) [19, 20]. Inspired by these successes, researchers have begun extending RL to the visual domain [21–25], including application in T2I generation [24, 25]. Notably, Group Relative Policy Optimization (GRPO)-based methods [20] have been applied to both diffusion models [26] and AR models [25]. However, MGMs remain largely unexplored in this context.

39th Conference on Neural Information Processing Systems (NeurIPS 2025).

Unlike AR and diffusion models, applying RL to MGMs poses a unique challenge: defining the transition probability in RL. In AR models, the transition probability corresponds naturally to next-token distributions. In diffusion models, each step's reverse denoising distribution acts as the transition probability. Since MGMs predict all masked tokens in parallel, a naive solution is to use the product of their probabilities at each iteration as the transition probability - effectively blending AR (token-based) and diffusion (iteration-based) perspectives. Yet, when applying it to our base model Show-o [27] using GRPO [20], as shown in Figure 1, we observe unsatisfactory performance.

This result prompts a deeper inquiry: *What is the correct or proper transition probability for MGMs?* Revisiting the core unmasking mechanism of MGMs, we find that it is the most confident tokens at each iteration, which we defined as newly unmasked tokens above, that matter most to the transition probability. Building on this, we propose two candidate definitions: (1) the product of probability over all newly unmasked tokens at each iteration. (2) the product of probability over both newly unmasked and remasked tokens, but treated with a probabilistic trick. We will elaborate on these formulations in Section 3.2.

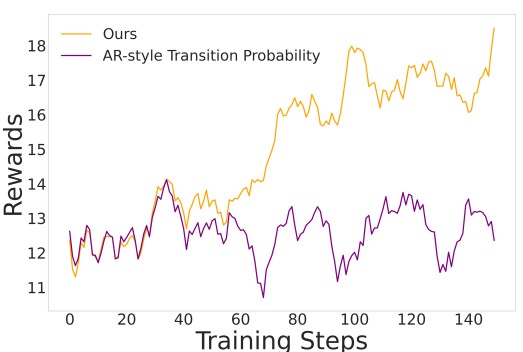

Figure 1: The unsatisfactory performance using the AR-style probability.

Building on this insight, we propose **Mask-GRPO**, the first approach that integrates RL into MGMs for T2I generation. Specifically, we model the unmasking process as a multi-step decision-making problem, using both transition possibility definitions we gave above. We then apply GRPO [20] into our base model, Show-o [27], resulting in a significant boosting performance. To further enhance Mask-GRPO, we explore the impact of removing the Kullback–Leibler (KL) constraint. We find that KL regularization is unnecessary if a sufficiently small learning rate is used.

However, RL requires efficient but high-quality sampling [28, 16]. This requirement presents two key challenges: (1) MGMs typically require a number of iterative steps to generate a final picture, while GRPO [20] requires revisiting all these steps, which is significantly time- and resource-consuming. (2) RL training is always unstable, which leads to less or no performance improvement. We term it as the 'Vanishing Samples' problem. On the one hand, it resembles the vanishing advantages problem [29] observed in LLMs; on the other hand, it differs from that because we mainly use CLIP [30] as our reward model, and it is almost impossible to obtain the same rewards within a group. It is more like a sample quality issue we will further discuss in Section 3.3.

To address these issues, we introduce a reduction strategy and a sample filtering strategy, respectively. The former reduces the iteration steps for computation or sample generation during training, and the latter uses a reward-based filtering mechanism to discard low-quality samples. Experiments support these strategies and lead to further improvements.

Our contributions are summarized as follows:

- We are the first to introduce RL to MGMs. We model the unmasking process as a multi-step decision-making problem and propose two effective formulations of the transition probability. Based on this, we develop Mask-GRPO, the first GRPO-based approach on MGMs for T2I generation.

- We explore several enhancements to push Mask-GRPO further, including removing the KL constraint, applying the reduction strategy, and filtering out low-quality samples. We analyze each and validate their effectiveness through extensive experiments.

- Our method achieves superior performance on standard T2I benchmarks and preference alignment, improving over the base model and matching or surpassing existing state-of-the-art approaches.

## 2 Preliminaries

### 2.1 Mechanism of Masked Generative Models

As summarized by [31], MGMs follow the encode-decode manner similar to AR models. In the encoding stage, an image is represented as a sequence of discrete tokens [32, 33], where each token corresponds to a categorical label. In the decoding stage, the model aims to recover the full token sequence and map it back to image pixels. However, unlike AR models, MGMs are trained to predict tokens in parallel using a bidirectional transformer, rather than token-by-token in an autoregressive manner. MGMs are always seen as a discrete diffusion process through the absorbing state ([MASK]), and we give it a detailed analysis in Appendix B.

Specifically, MGMs use an iterative decoding strategy over $T$ steps. Let $Y_t = [y^i]_{i=1}^N$ denote the latent token sequence at iteration $t$, where $N$ is the length of the reshaped token matrix. We denote $Y_t^M \subseteq Y_t$ and $Y_t^U \subseteq Y_t$ as the subset of all masked tokens and unmasked tokens, respectively. Note that

$$Y_t^M \cap Y_t^U = \emptyset, Y_t^M \cup Y_t^U = Y_t. \tag{1}$$

To generate an image, the model starts from a blank canvas where all tokens are masked:

$$Y_0^M = Y_0, Y_0^U = \emptyset, \tag{2}$$

and ends with a fully generated token sequence:

$$Y_T^M = \emptyset, Y_T^U = Y_T. \tag{3}$$

At each iteration $t$, the model predicts all masked tokens in $Y_t^M$ simultaneously and moves a subset of them, which we define as newly unmasked tokens in Section 1, to $Y_t^U$ to obtain $Y_{t+1}^U$, $Y_{t+1}^M$, and $Y_{t+1}$ for the next iteration. The number of tokens moved in each iteration is predefined as $n_t$, satisfying:

$$\sum_{t=0}^{T-1} n_t = N. \tag{4}$$

At each iteration $t$, the model first predicts the probabilities as $p_t \in \mathbb{R}^{N_t^M \times K}$, for all masked tokens in $Y_t^M$ in parallel, conditioned on all unmasked tokens in $Y_t^U$. Here, $K$ denotes the codebook size and $N_t^M$ is the size of $Y_t^M$, which means the number of masked tokens at iteration $t$. After that, for each mask token position $1 \leq i \leq N_t^M$, a token $y^i$ is sampled based on the prediction probabilities $p_t^i \in \mathbb{R}^K$, and a 'confidence' score is assigned based on this probability. $Y_{t+1}^U$ is then formed by choosing the most confident tokens from $Y_t^M$ and moving them to $Y_t^U$, according to their confidence scores. This Choose and Move (CaM) operation at each iteration $t$ can be formulated as:

$$CaM_t^i = \begin{cases} 1, & \text{if } cs_t^i > \text{sorted}(cs_t)[N_t^M - n_t] \\ 0, & \text{otherwise,} \end{cases} \tag{5}$$

where $cs_t^i$ is the confidence score for token $y^i$, and $cs_t$ is the list of all confidence scores at iteration $t$. Finally, the remaining tokens in $Y_t^M$, which we defined as newly remasked tokens in Section 1, are remasked to form $Y_{t+1}^M$ and subsequently, $Y_{t+1}$, for the next iteration.

### 2.2 Reinforcement Learning

RL is commonly formulated as a discounted Markov Decision Process (MDP) specified by the tuple $M = (S, A, P, R, \rho_0, \gamma)$, where $S$ and $A$ denote the state space and action space respectively, $P$ represents the transition dynamics, $R$ is the reward function, $\rho_0$ refers to the initial state distribution, and $\gamma$ denotes the discount factor. The agent interacts with the environment following a policy $\pi$, which generates a trajectory $\tau = (s_0, a_0, s_1, a_1, \cdots)$. The goal of RL is to find a policy that maximizes the expected return:

$$\pi^* = \arg\max_\pi E_{\tau \sim \pi}[\sum_{i=0}^{\infty} \gamma^i R(s_i, a_i)]. \tag{6}$$

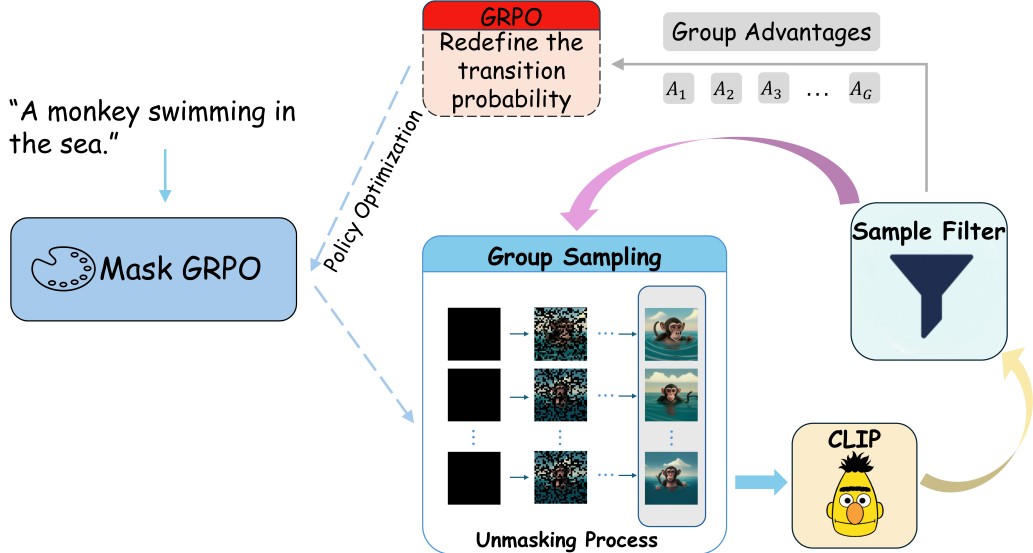

Figure 2: The framework of Mask-GRPO. We redefine the transition probability to formulate the unmasking process as a Markov decision-making problem. The reward is got from CLIP [30].

## 3 Methods

Our goal is to enhance MGMs for T2I generation using RL. To achieve this, we propose **Mask-GRPO**, the first RL approach specifically designed for MGMs, as illustrated in Figure 2. We begin by formulating the unmasking process in MGMs as a multi-step Markov decision-making problem in Section 3.1. Then, in Section 3.2, we propose two candidate definitions of transition probability building on our framework. Finally, we describe several enhancements for Mask-GRPO, including removing the KL constraint, applying a reduction strategy, and filtering out low-quality samples.

### 3.1 Unmasking as a Multi-Step Markov Decision-Making

Online RL improves policy performance through direct interaction with the environment, guided by reward signals. To integrate this with MGMs, we first formulate their unmasking process as a Markov decision-making problem.

Building on the mechanism in Section 2.1, we define the unmasking policy at each iteration $t$ as:

$$\pi_\theta(Y_{t+1}|Y_t, c), \tag{7}$$

where $c$ is the input prompt. This simplifies from the more explicit form:

$$\pi_\theta(Y_{t+1}^M, Y_{t+1}^U, Y_{t+1}|Y_t^M, Y_t^U, Y_t, c), \tag{8}$$

by leveraging the deterministic relationship between $Y_t^M/Y_t^U$ and $Y_t$ (as detailed in Equation (1)).

A crucial property for modeling a Markov decision-making problem is that each step must depend only on the current state and action. Fortunately, this holds in MGMs, where each iteration only depends on the current token sequence $Y_t$ and prompt $c$. Inspired by [26], we expand the $T$-iteration unmasking process into a $T$-step Markov decision process:

$$\begin{cases} s_t \triangleq (Y_t, c) \\ a_t \triangleq Y_{t+1} \\ p(s_{t+1}|s_t, a_t) \triangleq p_\theta(s_{t+1}|s_t, a_t) \\ \rho_0 \triangleq Y_0 \\ R(s_t, a_t) \triangleq \begin{cases} r(Y_t, c), & \text{if } t = T-1 \\ R(s_{T-1}, a_{T-1}), & \text{otherwise} \end{cases} \\ \gamma \triangleq \gamma, \end{cases} \tag{9}$$

where $0 \leq t \leq T-1$, $\gamma$ is a constant that will not be used in GRPO-based methods [20, 34], $r(Y_t, c)$ refers to the reward assigned to the final image generation result, and $p_\theta(s_{t+1}|s_t, a_t)$ denotes the transition probability we will discuss later. Based on this formulation, we integrate GRPO [20, 34] into MGMs and transform the learning objective from Equation (6) into the following:

$$J(\theta) = E_{c,\{s^j\}_{j=1}^G}$$

$$\left[\frac{1}{G}\frac{1}{T}\sum_{j=1}^{G}\sum_{t=0}^{T-1}\left(\min\left(r_t^j(\theta)A_t^j, \text{clip}\left(r_t^j(\theta), 1-\epsilon, 1+\epsilon\right)A_t^j\right) - \beta D_{KL}\left(\pi_\theta||\pi_{\text{ref}}\right)\right)\right],$$
(10)

where $G$ is the number of image generations within a group per prompt, and the $j$-th generation is presented by the superscript of $j$. The $\pi_{\text{ref}}$ denotes the reference policy (e.g., the base model). The likelihood ratio $r_t^j(\theta)$ and group-level normalized advantage $A_t^j$ are defined as follows, respectively:

$$r_t^j(\theta) = \frac{p_\theta(s_{t+1}^j|s_t^j, a_t^j)}{p_{\text{old}}(s_{t+1}^j|s_t^j, a_t^j)},$$
(11)

$$A_t^j = \frac{R(s_t^j, a_t^j) - \text{mean}(\{R(s_t^j, a_t^j)\}_{j=1}^G)}{\text{std}(\{R(s_t^j, a_t^j)\}_{i=1}^G)}.$$
(12)

So far, we have framed the unmasking process as a multi-step decision-making problem and incorporated GRPO into it. However, how to choose the reward function $r$ and model the transition probability $p_\theta(s_{t+1}|s_t, a_t)$ remains unknown. While we can use established perceptual models as our reward function (e.g., CLIP [30]), which we find effective, the second problem is more complex.

A straightforward idea is to mimic the autoregressive transition definition used in LLMs. For MGMs, since all tokens in $Y_t^M$ are predicted in parallel, we can define the AR-style transition probability as:

$$p_\theta(s_{t+1}|s_t, a_t) = \pi_\theta(Y_{t+1}|Y_t, c)$$
$$= \prod_{i \in Y_t^M} cs_t^i.$$
(13)

Here, $cs_t^i$ is the confidence score (i.e., predicted probability) for the $i$-th token at iteration $t$, as defined in Section 2.1. It can be understood as: AR models predict a token each time and take its token probability as the transition probability, while MGMs predict all tokens in $Y_t^M$ in parallel each iteration and take the product of every token probability as the transition probability.

While this definition is intuitive and consistent with AR models, our initial experiments reveal unsatisfactory performance when applying it directly to MGMs (as shown in Figure 1). This motivates a deeper investigation into whether this definition is inherently flawed — and if so, what a more principled alternative would be.

## 3.2 Two Candidate Definitions of Transition Probability

To address the limitations discussed above, we revisit the unmasking mechanism of MGMs and offer a key insight: The newly unmasked tokens (those with $CaM_t^i = 1$ in Equation (5)) play the most critical role in determining the transition probability. For simplicity here we denote $Y_t^{CaM} \subseteq Y_t^M$ as the subset of them.

To better illustrate this insight, we present an example in Figure 3 depicting the case of iteration $t = 0$. For simplicity, assume the image consists of only four tokens, each with a prediction distribution $p_t^i \in \mathbb{R}^K$, where $K = 4$ is the size of the codebook. We also assume that only one token needs to be predicted at this iteration (i.e., $n_t = 1$). Under this setup, we observe that

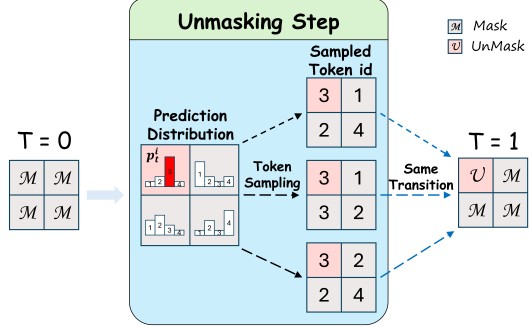

Figure 3: The illustration of transition probability. It is observed that different token samplings finally can lead to the same next state.

multiple different token samplings and confidence scores (i.e., predicted probabilities) can lead to the same transition $(s_{t+1}, s_t, a_t)$, provided the following conditions hold:

- The sampling for the newly unmasked tokens $Y_t^{CaM}$ remains the same.
- The remaining tokens in $Y_t^M \backslash Y_t^{CaM}$, which will be remasked, have lower confidence scores $cs_t^i$ than any of newly unmasked tokens in $Y_t^{CaM}$.

These lead us to define the transition probability $p_\theta(s_{t+1}|s_t, a_t)$ by explicitly modeling these two factors. Let $\min(cs_t)$ be the minimum confidence score among $Y_t^{CaM}$. Then, our first candidate definition of transition probability can be defined as:

$$p_\theta(s_{t+1}|s_t, a_t) = \left( \prod_{i \in Y_t^{CaM}} cs_t^i \right) \cdot \left( \prod_{i \in \left( Y_t^M \backslash Y_t^{CaM} \right)} \left( \sum_{k:\left(p_t^i\right)^k < \min(cs_t)} \left(p_t^i\right)^k \right) \right). \quad (14)$$

Here, $(p_t^i)^k$ refers to the probability of sampling the $k$-th token at $i$-th token position at iteration $t$, which is the $k$-th column in $p_t^i$. While Equation (14) provides a mathematically grounded definition of transition probability, we also propose a second, empirically motivated definition:

$$p_\theta(s_{t+1}|s_t, a_t) = \left( \prod_{i \in Y_t^{CaM}} cs_t^i \right). \quad (15)$$

We refer to the definition in Equation (14) and Equation (15) as $p_{\theta 1}$ and $p_{\theta 2}$, respectively. Note that $p_{\theta 2}$ is a simplification of $p_{\theta 1}$, focusing solely on the newly unmasked tokens $Y_t^{CaM}$ and ignoring others. Despite this simplification, both definitions proved effective in our experiments, with $p_{\theta 1}$ offering greater theoretical rigor and $p_{\theta 2}$ offering computational simplicity.

With these definitions in place, we now possess all the essential components needed to apply RL to MGMs. In the next subsection, we introduce several strategies to further enhance the performance of our proposed method.

### 3.3 Enhancing Strategies

Building upon our framework in Section 3.1 and the two candidate transition probabilities in Section 3.2, we introduce several practical strategies to enhance performance. These include: removing the KL constraint, applying a reduction strategy, and filtering out low-quality samples.

**Removing Kullback–Leibler Constraint.** Inspired by recent advances in LLMs [29, 35], we experiment with removing the KL constraint, setting $\beta = 0$ in Equation (10). The motivation lies in the size of our base model — approximately 1.3 billion parameters — which is relatively small. For such models, the KL constraint may hinder explorations. As demonstrated in Section 4.3, removing the KL term improves the generation performance. Note that this somewhat contrasts with the findings in [36], where the KL term was shown to be beneficial. We attribute this discrepancy to the difference in model size: our model uses the 1.3B Show-o [27], while theirs employs the 2.5B Stable Diffusion 3.5 Medium (SD3.5-M) [5]. This aligns with trends observed in LLMs, where larger models benefit from KL regularization, whereas smaller models may suffer from it.

**Reduction Strategy.** As denoted in Section 1, image synthesis in MGMs requires multiple unmasking iterations. While MGMs are more efficient than standard AR models or diffusion models, revisiting every step to compute the objective in Equation (10) introduces significant computational overhead. Inspired by [37], which shows that even the first diffusion step captures key trajectory trends, we introduce the two following reduction strategies:

1. Computational reduction strategy, which computes the object in Equation (10) over only a subset of the total iterations (e.g., the first or last 25 out of 50 steps).

2. Unmasking reduction strategy, which reduces total number of unmasking iterations during training (e.g., from $T = 50$ to $T = 20$), while maintaining the full unmasking schedule during evaluation.

The computational reduction strategy aims to prevent revisiting all iterations, while the unmasking reduction strategy focuses on reducing the total number of unmasking iterations. As shown in Section 4.3, the second strategy accelerates training while only suffering slight performance degradation.

Table 1: Model Comparison on GenEval.

| Model | Params. | Single Obj. | Two Obj. | Counting | Colors | Position | Color Attri. | Overall↑ |
|---|---|---|---|---|---|---|---|---|
| **Diffusion or Flow Matching Models** | | | | | | | | |
| PixArt-alpha [38] | 0.6B | 0.98 | 0.50 | 0.44 | 0.80 | 0.08 | 0.07 | 0.32 |
| SD1.5 [39] | 0.9B | 0.97 | 0.38 | 0.35 | 0.76 | 0.04 | 0.06 | 0.43 |
| SD2.1 [39] | 0.9B | 0.98 | 0.51 | 0.44 | 0.85 | 0.07 | 0.17 | 0.50 |
| LDM [39] | 1.4B | 0.92 | 0.29 | 0.23 | 0.70 | 0.02 | 0.05 | 0.37 |
| SD-XL [40] | 2.6B | 0.98 | 0.74 | 0.39 | 0.85 | 0.15 | 0.23 | 0.55 |
| DALLE-2 [41] | 6.5B | 0.94 | 0.66 | 0.49 | 0.77 | 0.10 | 0.19 | 0.52 |
| SD3.5-L [5] | 8B | 0.98 | 0.89 | 0.73 | 0.83 | 0.34 | 0.47 | 0.71 |
| DALLE-3 [3] | - | - | - | - | - | - | - | 0.67 |
| **Autoregressive Models** | | | | | | | | |
| LlamaGen [8] | 0.8B | 0.32 | 0.71 | 0.34 | 0.21 | 0.58 | 0.07 | 0.04 |
| JanusFlow [42] | 1.3B | 0.97 | 0.59 | 0.45 | 0.83 | 0.53 | 0.42 | 0.63 |
| Janus [43] | 1.3B | 0.97 | 0.68 | 0.30 | 0.84 | 0.46 | 0.42 | 0.61 |
| SimpleAR-1.5B [25] | 1.5B | - | 0.90 | - | - | 0.28 | 0.45 | 0.63 |
| Infinity (+Rewriter) [44] | 2B | - | 0.85 | - | - | 0.49 | 0.57 | 0.73 |
| Chameleon [45] | 7B | - | - | - | - | - | - | 0.39 |
| Emu3 (+Rewriter) [46] | 8.5B | 0.99 | 0.81 | 0.42 | 0.80 | 0.49 | 0.45 | 0.66 |
| **Masked Generative Models** | | | | | | | | |
| MaskGen-XL [11] | 1.1B | 0.99 | 0.61 | 0.55 | 0.81 | 0.13 | 0.31 | 0.57 |
| Meissonic [12] | - | 0.99 | 0.66 | 0.42 | 0.86 | 0.10 | 0.22 | 0.54 |
| Show-o [27] | 1.3B | 0.95 | 0.52 | 0.49 | 0.82 | 0.11 | 0.28 | 0.53 |
| **Show-o+Mask-GRPO** | 1.3B | **0.99** | **0.90** | **0.69** | **0.85** | **0.35** | **0.59** | **0.73** |

[1] Results for models other than our approach are from [47] or their original papers.

**Sample Filtering.** During RL training, we observed frequent instability: reward scores often degrade temporarily, and the performance of the final model stagnates. We identify this critical limitation as 'Vanishing Samples'. We notice that similar trends have been concurrently observed in [29, 21], and they are attributed to equal rewards within the group of samples, resulting in:

$$R^j - \text{mean}(\{R^j\}_{j=1}^G) = 0. \tag{16}$$

This leads to zero advantages $A$ in Equation (12). However, our setting differs from theirs in that we use a CLIP-based reward model, not a rule-based one. Therefore, exact reward duplication within a group is unlikely. Instead, it is more like a sample quality issue, where important positive samples within a group become indistinguishable due to low standard deviation in rewards. This is particularly problematic when the reward model fails to accurately distinguish good from bad generations.

To mitigate this, we propose a dynamic sampling strategy that filters out low-quality samples. To be specific, we set a dynamic filtering threshold based on the history standard deviation of the rewards within each group. If a newly generated group has a reward standard deviation below the threshold, then we resample the group. In practice, we set it as the lowest 10th percentile of the history standard deviation.

To summarize above, we present the framework of our Mask-GRPO with sample filtering and removing the KL constraint, which achieves the best performance in our experiments. In the following sections, the default Mask-GRPO applies the $p_{\theta 1}$ defined in Equation (14), employs sample filtering, removes the KL constraint, and retains the full iteration steps during both training and evaluation.

Table 2: Model Comparison on MSCOCO 30K FID.

| Model | Params. | FID-30K↓ |
|---|---|---|
| PixArt [38] | 0.6B | 7.32 |
| GigaGan [48] | 0.9B | 9.09 |
| SD1.5 [39] | 0.9B | 9.62 |
| Janus [43] | 1.3B | 7.11 |
| LDM [39] | 1.4B | 12.64 |
| DALLE-2 [41] | 6.5B | 10.39 |
| DreamLLM [49] | 7B | 8.76 |
| LWM [50] | 7B | 12.68 |
| DALL.E [51] | 12B | 27.50 |
| SEED-X [52] | 17B | 14.99 |
| Show-o [27] | 1.3B | 9.24 |
| **Show-o+Mask-GRPO** | 1.3B | **8.32** |

# 4 Experiments

## 4.1 Experiment Setup

**Training Settings.** We conduct our experiments using the training set of LAION dataset [53] without accompanying images. Inspired by [25], we select prompts of varying lengths — both short and long — as they have been shown to provide more informative signals for reinforcement learning. The base model we selected is Show-o [27]. We employ CLIP [30] as the primary reward model. In ablation studies Section 4.3, we additionally validate our approach with ImageReward [54]. Additional implementation details are provided in Appendix C.

**Evaluation Details.** Following Stable Diffusion [39, 5], we evaluate MASK-GRPO with standard T2I benchmarks GenEval [55] and FID [56] on the MSCOCO dataset [57]. In ablation studies Section 4.3, we additionally evaluate our method with ImageReward [54] for preference alignment. Notably, our training set does not contain any GenEval-style prompts. As a result, the GenEval [55] prompts serve as zero-shot evaluations, thereby providing a more robust assessment of generalization performance.

## 4.2 Main Results

Table 1 presents the GenEval results, while Table 2 reports the MSCOCO-30K FID score for our proposed Mask-GRPO. It can be observed that our approach significantly enhances the T2I capabilities of the base model, with an improvement in GenEval to 0.73 and MSCOCO-30K FID to 8.32. These validate the effectiveness of Mask-GRPO. Qualitative comparisons are illustrated in Figure 4, with additional visualization results included in Appendix D.

Remarkably, Mask-GRPO outperforms all existing state-of-the-art methods on GenEval [55], despite using a smaller base model (1.3B) than most competing approaches. Moreover, Mask-GRPO shows strong zero-shot generalization, as GenEval [55] is evaluated under zero-shot settings in our experiments. This also highlights the effectiveness of employing CLIP as the reward model in our framework. A further discussion on CLIP as the reward model is provided in Section 4.4.

Moreover, we find that both of $p_{\theta 1}$ and $p_{\theta 2}$ are useful, with the former converging faster.

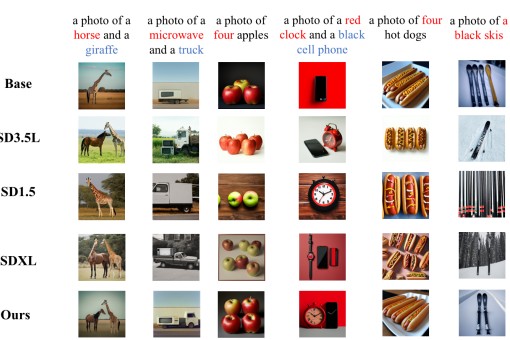

Figure 4: Qualitative Comparisons on GenEval.

## 4.3 Ablation Studies

The results of all ablation studies are presented in Table 3.

**Removing KL.** It can be observed that removing the KL term significantly boosts performance, aligning with our hypothesis that KL may hinder exploration for small models. However, we also observe that the absence of the KL term makes Mask-GRPO more sensitive to hyperparameters, particularly the learning rate. A more detailed analysis is provided in Appendix C.

**Reduction Strategy.** Among the two reduction strategies, the unmasking approach proves to be more effective, achieving a better balance between computational efficiency and performance. Although it entails a slight performance drop, it reduces computational requirements by approximately 3–4×, making it a highly practical choice for RL fine-tuning of large models.

Table 3: Ablation Results.

| Method | GenEval ↑ |
|---|---|
| Show-o | 0.53 |
| + Mask-GRPO | 0.73 |
| + Mask-GRPO w/ KL | 0.62 |
| + Mask-GRPO w/o sample filter | 0.60 |
| + Mask-GRPO w/ Computation Reduction Strategy | 0.56 |
| + Mask-GRPO w/ Unmasking Reduction Strategy | 0.66 |

**Sample Filtering.** The results indicate that filtering out low-quality samples improves the final performance, supporting a key insight that data quality plays a crucial role in reinforcement learning

for text-to-image generation. Unlike LLMs, where curated RL fine-tuning datasets are increasingly available, the T2I domain currently lacks specialized datasets for RL fine-tuning. We hope that this gap will be addressed in future work to further advance the field.

**Reward Model.** Finally, we test Mask-GRPO's robustness under different reward models by replacing CLIP with ImageReward [54] as the reward model. Results in Table 4 confirm that Mask-GRPO consistently enhances the T2I generation performance across different reward models, thereby validating its generality.

Table 4: Results with ImageReward [54].

| Model | ImageReward ↑ |
|---|---|
| Show-o | 1.02 |
| Show-o + ImageReward | 1.23 |

Table 5: Additional Experiments on Relative Correctness.

| Original Prompts | $\frac{N}{2}$ | N | 2N |
|---|---|---|---|
| 'a photo of six clocks' | 32.5298 | 35.8555 | 33.2857 |
| 'a photo of four handbags' | 29.7800 | 33.0426 | 32.4762 |
| 'a photo of two backpacks' | 28.4991 | 34.1256 | 29.7021 |
| 'a photo of two frisbees' | 32.1401 | 36.8218 | 29.6494 |
| 'a photo of four toothbrushs' | 33.8095 | 38.2953 | 34.8197 |
| 'a photo of four vases' | 28.2428 | 37.1507 | 29.9521 |
| 'a photo of two computer keyboards' | 32.3883 | 37.2445 | 31.1868 |
| 'a photo of four baseball gloves' | 29.5035 | 33.6917 | 31.3151 |
| 'a photo of two beds' | 33.5974 | 38.9766 | 31.7446 |
| 'a photo of four giraffes' | 29.2052 | 36.3132 | 32.4882 |
| **Overall** | **30.9696** | **35.1873** | **31.6620** |

## 4.4 Further Discuss

In this subsection, we further discuss the underlying reward mechanisms of CLIP when used as the reward model. Although CLIP is known to have limitations in counting and attribute binding, we nevertheless observe notable improvements in these aspects, as shown in Table 1.

We believe the success of our method stems from the design of GRPO, where we do not rely on absolute correctness, but rather on relative correctness within a sample group. In GRPO-based methods, we do not use raw rewards from the reward model directly. Instead, they are transformed into group-level normalized advantage (as shown in Equation (12)), which effectively compares the relative correctness among samples within each group. Therefore, even if CLIP cannot provide an accurate 'absolute correctness' for counting or attribute correctness, as long as it captures the relative trend, it can still drive the model to improve.

To verify this, we conduct an additional experiment. We randomly select 10 GenEval-counting-style prompts (e.g. 'a photo of $N$ items') and generate 10 correct images accordingly. We then modify each prompt to 'a photo of $\frac{N}{2}$ items' and 'a photo of $2N$ items', while keeping the originally generated (correct) images. We computed the CLIP-scores for each prompt-image pair. As shown in Table 5, CLIP consistently assigns higher scores to prompts matching the correct count, demonstrating that it can indeed capture 'relative correctness'. This justifies the use of CLIP within our framework and explains why Mask-GRPO can improve performance in counting and attribute-binding tasks.

## 5 Conclusion and Future Work

In this paper, we present Mask-GRPO, a GRPO-based approach to improve MGMs in T2I generation. It is the first to introduce online RL to MGMs. We revisit the definition of transition probability meticulously, finding that the corresponding definitions in AR and diffusion models are not feasible for MGMs. By redefining two candidate transition possibilities, we successfully integrate GRPO [20] into MGMs. Consequently, we explore several useful strategies to further improve our method.

While achieving superior results, Mask-GRPO's potential is still not fully revealed. On the one hand, applying such methods on larger base models to explore possible scaling-up will be interesting work.

On the other hand, Mask-GRPO's potential on video generation is valuable, while text-to-video generation is always more complicated as it has to generate coherent outputs that are both contextually and temporally consistent.

The reward model remains a key challenge for T2I generation. First, unlike RL post-training in LLMs, the application of RL to T2I generation depends on effective reward models for evaluating generated images. In this work, we adopt CLIP as the reward model, which proves effective. However, similar to other reward models, CLIP is still narrow in scope and not sufficiently accurate to support further improvements in RL post-training. Developing more robust and fair reward models, or jointly training them during RL, may help mitigate these limitations.

## Acknowledgments and Disclosure of Funding

This work was supported in part by the Natural Science Foundation of Shenzhen (No. JCYJ202308 07111604008, No. JCYJ20240813112007010), the Natural Science Foundation of Guangdong Province (No. 2024A1515010003), National Key Research and Development Program of China (No. 2022YFB4701400) and Cross-disciplinary Fund for Research and Innovation of Tsinghua SIGS (No. JC2024002).

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

# A    Related Work

## A.1    Reinforcement Learning in Large Language Models

RL [16–18, 58] has emerged as a pivotal paradigm for refining LLMs [19, 20, 59–62]. Prominent approaches like Reinforcement Learning from Human Feedback (RLHF) [63] and Direct Preference Optimization (DPO) [64] have demonstrated remarkable success in enhancing the safety and instruction-following capabilities of LLMs [63, 65] and multimodal LLMs (MLLMs) [66, 67]. Recently, the emergence of OpenAI o1 [19] and DeepSeek-R1 [20] has sparked renewed interest in incentivizing reasoning capabilities in LLMs via RL. Notably, the GRPO algorithm [34] utilizes the rule-based reward design, demonstrating the huge potential for large-scale RL applications on LLMs. Inspired by this, recent works aim to improve the original GRPO in LLMs [29] or attempt to apply it to MLLMs [68–70].

## A.2    Visual Generation Models

**Diffusion Models.** In recent years, numerous diffusion-based methods [1, 6, 39, 41, 40, 38] have demonstrated exceptional capabilities in T2I generation. In this framework, the model is tasked with predicting the added Gaussian noise. Recently, flow matching [71, 72] follows the core idea of diffusion models but treats generation as learning a continuous-time normalizing flow, achieving competitive visual generation results with fewer denoising steps [5]. However, these methods still suffer from their unstable controllability and low inference speed due to the multi-denoising steps.

**Autoregressive Models.** The transformer models with autoregressive output schemes have demonstrated great success in modeling both language and multi-modality generation [73–77]. Inspired by such progress, a series of works [78, 32, 51, 45] utilize such autoregressive paradigms with causal attention to learn the dependency of image pixels for image generation. More recently, [8] also demonstrated that images can be generated through the LLMs' architecture in an autoregressive way. However, such raster-order autoregression suffers from high computational cost and performance bottlenecks when synthesizing high-resolution and high-fidelity images. [79].

**Masked Generative Models.** To address the challenges of diffusion models and standard AR models, MaskGiT [10] first introduces a new bidirectional transformer for image synthesis modeling image generation as a mask prediction problem. During training, it is trained on a similar proxy task to the mask prediction in Bert [15], while at inference time it adopts a novel paralleled decoding method to synthesize an image in a constant number of steps. Inspired by its success, more MGMs [14, 80] start to emerge and have gained significant success in T2I generation. Recently, this approach has been effectively extended by Show-o [27] and MAR [81], in the aspect of visual understanding-generation unification and continuous-valued tokens integration. Considering the efficiency and recent progress of MGMs, it is worthwhile to further improve their performance by RL.

## A.3    Reinforcement Learning in Text-to-Image Generation

The success of RL for LLMs also incentivizes research to try RL on T2I generation. Prior works mainly focus on aligning pretrained T2I models with human preferences [82–85], improving aesthetics and semantic consistency. Very recently, inspired by GRPO [20], new approaches have been proposed trying to apply GRPO-based methods on T2I generation [86, 36, 87]. However, all these works are done in diffusion-based models or standard AR models, overlooking the masked generative models. In this paper, we are trying to fill this gap.

# B    Masked Generative Models as Discrete Diffusion Process

The theory of discrete diffusion in generative models focuses on applying a Markov process to corrupt data in discrete states, followed by a reverse process for reconstruction. The forward process $q(x_t|x_{t-1})$ is represented by a categorical distribution:

$$q(x_t|x_{t-1}) = \text{Cat}(x_t|x_{t-1}Q_t),$$

where $Q_t$ is the transition matrix, and the corruption process can be tailored with different structured matrices, such as uniform or absorbing state models.

## B.1 Forward Process

The forward process gradually corrupts data by replacing each token with either a new token or an absorbing state (like a $[MASK]$ token). For discrete tokens, the transition matrix $Q_t$ is formed by a product of matrices over time $Q_t = Q_1 Q_2 \ldots Q_t$. The forward process can be mathematically expressed as:

$$q(x_t|x_0) = \text{Cat}(x_t|x_0 Q_t),$$

where $Q_t$ may represent structured transition patterns like Gaussian kernels, nearest-neighbor mappings, or absorbing states.

## B.2 Reverse Process

The reverse process $p_\theta(x_{t-1}|x_t)$ is learned to denoise the corrupted data, starting from a noisy state $x_T$ and moving towards $x_0$. The reverse process can be parameterized by a neural network to predict the most likely previous state given the current noisy state:

$$p_\theta(x_{t-1}|x_t) = \sum_{x_0} q(x_{t-1}|x_t, x_0) p_\theta(x_0|x_t).$$

This reverse process learns to undo the corruption step-by-step, ideally predicting the original data distribution.

## B.3 Structured Denoising Diffusion in Discrete State Spaces

In **Discrete Denoising Diffusion Probabilistic Models (D3PM)**, the forward process is generalized by incorporating structured transition matrices, allowing for richer and more controlled data corruption processes. These models do not require embedding continuous data into latent continuous spaces but instead operate entirely in discrete spaces.

### B.3.1 Transition Matrices for Discrete Diffusion

The choice of the transition matrix $Q_t$ plays a crucial role in the quality of the model. There are several variants, such as:

- **Uniform Transition**: This is a simple case where each state has an equal chance of transitioning to any other state.
- **Absorbing State Models**: Here, a token may either stay the same or transition into an absorbing state (e.g., $[MASK]$).
- **Discretized Gaussian**: For ordinal data like images, the transition probabilities are biased toward states that are closer in terms of similarity or embedding space.

The corruption process is defined in Appendix B.1, where the transition matrix $Q_t$ is constructed from domain knowledge or data-specific structure. The reverse process aims to recover the original data $x_0$ using a learned model.

## B.4 Variational Loss Function for D3PM

To train D3PM models, a variational loss function is used, which combines both the log-likelihood of the observed data and the KL divergence between the forward and reverse distributions. The loss function for training D3PMs is expressed as:

$$L_{\text{ELBO}} = \mathbb{E}_{q(x_0)} \left[ \sum_{t=1}^{T} D_{\text{KL}}[q(x_t|x_0) \parallel p_\theta(x_t)] \right].$$

For the discrete case, the transition matrices and their structure determine how the data is corrupted and reconstructed, and the objective is to minimize the difference between the corrupted and reconstructed data over all steps.

## C   Implementation Details

All experiments are conducted on 16 NVIDIA A100 80GB GPUs. The learning rate is set as $3e - 6$ and the group size $G$ set as is 6. We utilize Adam as our optimizer and set beta to 0.95, and we set the batch size to 96 (6 rollouts per GPU for one prompt).

During training, we find that it is more sensitive when we remove the KL divergence constraint. Specifically, when we use the KL constraint, we can set the learning rate as $5e - 6$, while at most $3e - 6$ when removing the KL divergence constraint. The training will be collapsed if we set a learning rate of $5e - 6$ and remove the KL constraint at the same time.

## D   Visualization

Some image generation results are shown in Figure 5. The corresponding prompts are (each line, from left to right):

Line 1:

- Create an image of a cat as a gardener, wearing a straw hat, gardening gloves, and surrounded by colorful flowers.
- Anime illustration of Princess Mononoke from Studio Ghibli, by artgerm, stunning artwork.
- Futuristic cyberpunk city at night, neon lights, high-tech car, vibrant colors, cinematic lighting, highly detailed, sci-fi atmosphere, 8k resolution, unreal engine.
- A whimsical pink cloud-shaped building with minimalist windows and doors, floating above a vibrant blue sky with cotton-like clouds, Studio Ghibli-style animation movie texture.

Line 2:

- A white puppy sitting playfully in autumn leaves, surrounded by fallen red apples, soft natural lighting.
- Heroic elf warrior, golden glowing background, detailed fantasy armor, cinematic lighting, epic fantasy art, high detail.
- Vibrant city skyline during sunset, modern skyscrapers, colorful abstract style, warm gradient sky, digital art, urban landscape, vivid colors.
- Mystical forest with glowing mushrooms and a babbling brook.

Line 3:

- Futuristic metallic humanoid robot, highly detailed face, sci-fi background, cinematic lighting, dystopian cityscape, 4K resolution.
- A cute duck wearing a chef uniform covered in cookie batter, unreal engine render 8k.
- A realistic Venusaur animal among the trees, forest lake, moss, cold weather, dark teal and amber, Sony A7 IV.
- 4d photographic image of full body image of a cute little chibi boy realistic, vivid colors octane render trending on artstation, artistic photography, photorealistic concept art, soft natural volumetric cinematic perfect light, UHD no background.

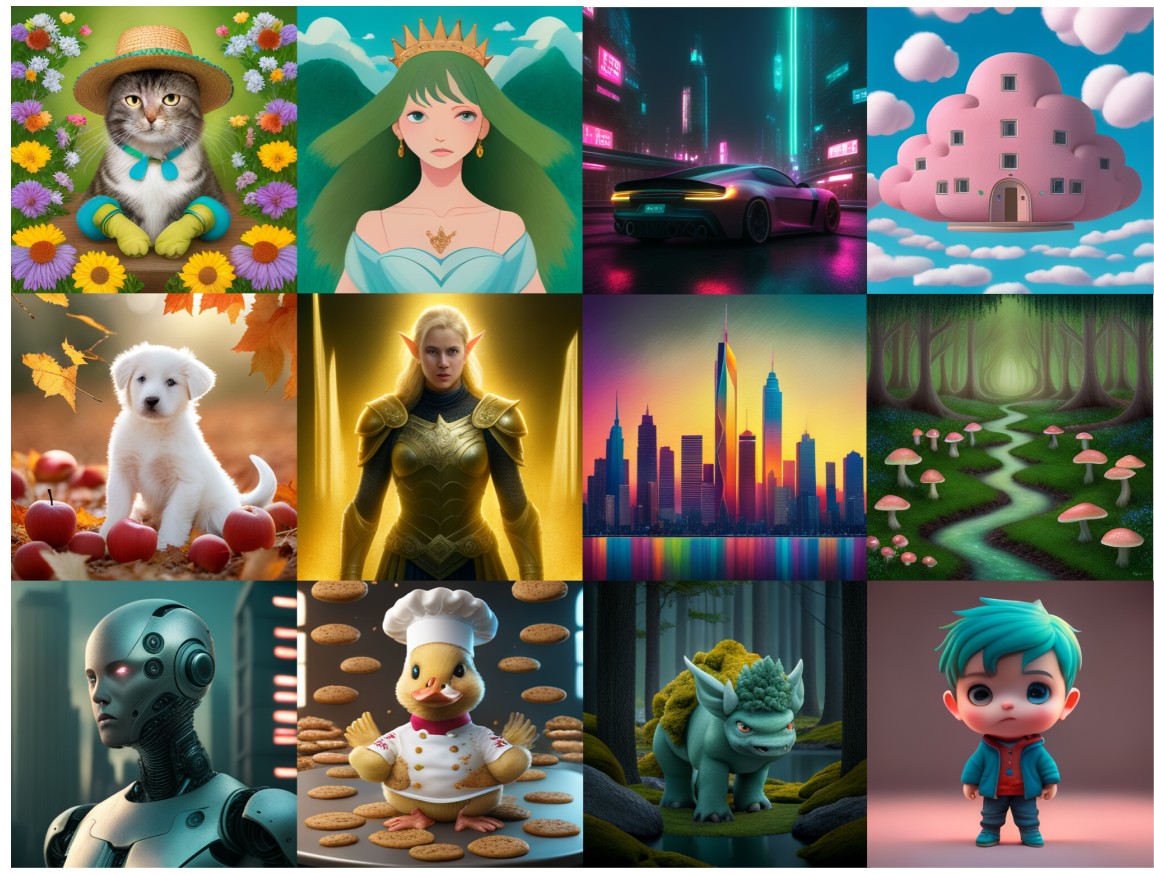

Figure 5: Visualization results of Mask-GRPO.

