# OpenReview forum: "Reinforcement Learning Meets Masked Generative Models: Mask-GRPO for Text-to-Image Generation"
_NeurIPS.cc/2025/Conference — NeurIPS 2025 poster_

### Official Review · Reviewer_5UPe · 2025-06-03

**Clarity:** 2
**Significance:** 2
**Originality:** 2
**Rating:** 3
**Confidence:** 4

**Summary:**

This paper introduces **Mask-GRPO**, a novel approach that integrates online reinforcement learning (RL) into the underexplored paradigm of **Masked Generative Models (MGMs)** for text-to-image (T2I) generation. While prior RL-based methods have focused primarily on diffusion and autoregressive models, this work is the first to apply **Group Relative Policy Optimization (GRPO)** to MGMs. The authors reformulate the token unmasking process in MGMs as a multi-step Markov decision-making problem and propose two new definitions of the transition probability to better align with the unique structure of MGMs. To further enhance training efficiency and stability, they introduce several strategies including removing the Kullback–Leibler (KL) constraint, reducing the number of unmasking iterations, and filtering out low-quality samples based on reward variance. Experimental results on GenEval and MSCOCO benchmarks show that Mask-GRPO significantly improves over the base model, achieving 38% and 10% gains respectively, and demonstrates strong zero-shot generalization. These findings highlight the potential of applying online RL to MGMs, especially in settings with limited model size and data.

**Questions:**

1. **Clarification on Transition Probability Formulations (Eq. 14 vs. Eq. 15)**
Could the authors provide further intuition and possibly a visual explanation for the two proposed transition probability definitions?
In particular, what are the implications of including the re-masked token terms in Eq. 14 versus excluding them in Eq. 15 in terms of bias/variance or convergence behavior?
A more detailed comparison in both theory and experiment would help justify the final choice of `pθ₁`.

2. **Robustness to Reward Models**
The current implementation heavily depends on CLIP for reward evaluation. Have the authors considered alternative reward models (e.g., human feedback, aesthetic scoring models, or task-specific metrics)?
It would be valuable to understand whether Mask-GRPO's performance holds under different reward functions or if it's highly tuned to CLIP.
Including at least one additional reward model in the ablation would strengthen the generality claim.

3. **Generalization to Other MGMs**
The paper uses Show-o as the sole base model. Have the authors tested Mask-GRPO on other MGM variants (e.g., MaskGIT or Meissonic)?
Results on a second MGM would reinforce that the method is not overfitted to a specific architecture or training schedule.

4. **Sample Efficiency and Compute Costs**
The method introduces strategies like iteration reduction and sample filtering to reduce overhead, but a more detailed analysis of the compute/resource trade-off would be helpful.
For example: how many GPU-hours were saved by unmasking reduction vs. how much performance was sacrificed?

5. **Theoretical Justification or Guarantees**
Although the GRPO framework is well-established, could the authors elaborate on why it is particularly well-suited for the masked generation setting?
Are there theoretical intuitions or constraints under which GRPO is expected to perform better than PPO or DPO in MGMs?

---

**Ethical Concerns:**

["NO or VERY MINOR ethics concerns only"]

**Final Justification:**

After carefully rereading the manuscript, I find that the authors’ rebuttal provides a clearer explanation than the paper itself. This leaves me with mixed impressions: while the method may be solid, the submitted manuscript is largely ineffective at conveying its ideas. Nonetheless, my overall rating is now somewhat more positive.

**Limitations:**

**No.**

While the paper is technically thorough and presents strong empirical results, it does **not sufficiently discuss the limitations or potential societal impacts** of applying reinforcement learning to text-to-image generation using masked generative models. Some constructive suggestions for improvement:

The authors could explicitly address the following:

   * The **reliance on CLIP** as a reward model, which may encode biases or misalignments with human judgment.
   * The **scalability** of Mask-GRPO to larger models or datasets, particularly in terms of compute and memory cost.
   * The potential **overfitting** or reward hacking issues in long training runs, which are common in RL-based systems.

**Paper Formatting Concerns:**

I do not find.

**Quality:**

2

**Strengths And Weaknesses:**

### **Strengths**


  The paper makes a novel and timely contribution by introducing the first reinforcement learning (RL) method tailored for **Masked Generative Models (MGMs)** in text-to-image generation. Most prior work in this area has been focused on diffusion and autoregressive models. The proposal to apply **Group Relative Policy Optimization (GRPO)** in this new setting is both original and well-motivated.


  The methodology is sound and well-grounded. The authors carefully reformulate the unmasking process of MGMs as a multi-step Markov decision process and propose two principled definitions of transition probability, demonstrating a deep understanding of the underlying generative mechanism. Enhancements like KL removal, iteration reduction, and sample filtering are empirically validated and contribute to practical robustness.


  The proposed Mask-GRPO achieves strong empirical performance, outperforming a strong MGM baseline (Show-o) by significant margins: 38% on GenEval and 10% on MSCOCO FID. These improvements are notable given the small model size (1.3B parameters), suggesting high sample efficiency and scalability potential. This work opens the door for further RL applications in efficient, non-autoregressive generative frameworks.


  The paper is generally well-structured, with clear exposition of the motivations, methodology, and experimental design. Key ideas, such as the redefinition of transition probability and the rationale for each enhancement strategy, are communicated effectively with supporting figures and ablations.

---

### **Weaknesses**


  While the main ideas are well-explained, some technical details—especially regarding the exact computation of transition probabilities and sampling logic in the proposed definitions—could benefit from further clarification or simplification. Equations (14) and (15), while correct, may be difficult to parse without strong background knowledge.


  The paper lacks a deeper theoretical analysis of the convergence or stability properties of Mask-GRPO in the MGM setting. While empirical evidence supports its effectiveness, a theoretical justification for why the new transition probability definitions lead to better learning would further strengthen the work.


  The evaluation is conducted on a single base model (Show-o) and specific benchmarks (GenEval and MSCOCO FID). It is unclear how generalizable the method is across different masked generative models or reward functions beyond CLIP-based scoring.


  The paper relies heavily on CLIP as the reward model, but does not explore or compare alternative reward formulations. This limits understanding of how robust or sensitive the method is to different types of reward signals.

---

---

> ### Author Rebuttal · Authors · 2025-07-31
>
> Dear Reviewer 5UPe,
>
> Thank you for your time and thoughtful feedback on Mask-GRPO. We are pleased that you recognize the novelty and effectiveness of our method. Please find our responses below:
>
> # Q1: Clarification on Transition Probability Formulations
> > Could the authors provide further $\cdots$ justify the final choice of $p_{\theta 1}$.
>
> Thanks for your valuable question. Due to the new regulation this year, we are not allowed to provide any visual contents in rebuttal. However, we are happy to elaborate a more in-depth discussion of $p_{\theta 1}$ and $p_{\theta 2}$, in Equation (14) and Equation (15), respectively.
>
> In brief, $p_{\theta 2}$ is a simplification of $p_{\theta 1}$, focusing solely on the newly unmasked tokens and ignoring others. While $p_{\theta 1}$ offers greater theoretical rigor (real transition probability) and lower bias in theory, we found them both effective in our experiments, resulting almost the same GenEval performance, as shown in Table 1 (in the paper).
>
> A slight advantage $p_{\theta 1}$ has over $p_{\theta 2}$ is its robustness to the learning rate. While $p_{\theta 2}$ focuses solely on the newly unmasked tokens, it is more sensitive to the learning rate. The default learning rate we use is 3e-6 for $p_{\theta 1}$, and 2e-6 for $p_{\theta 2}$, respectively.
>
> Thanks for your question. We hope we explain well under the visual content limitation.
>
> # Q2: Robustness to Reward Models
> > The current implementation heavily depends on CLIP $\cdots$ strengthen the generality claim.
>
> Thanks for your valuable suggestion. We have conducted an additional experiment with two more reward models, ImageReward[1] and UnifiedReward[2]. We follow the same setting in default Mask-GRPO and the results on GenEval are shown below:
>
> **Table 8.** Performence of Various Reward Models.
> | Reward Model | Geneval Score |
> | ----- | ------ |
> | Baseline | 0.53 |
> | ClIP | 0.73 |
> | ImageReward | 0.70 |
> | UnifiedReward | 0.65 |
>
> As shown, both ImageReward and UnifiedReward significantly outperform the baseline (by 32% and 23%, respectively), demonstrating the adaptability and effectiveness of Mask-GRPO across different reward models. Importantly, these results were achieved without any fine-tuning of hyperparameters, indicating potential for even higher performance with model-specific tuning.
>
> We appreciate your suggestion and will include these results in the revised manuscript.
>
> # Q3: Generalization to Other MGMs
> > The paper uses Show-o  $\cdots$ specific architecture or training schedule.
>
> Thanks for your question. Actually, the decoding strategy of our base mdoel, Show-o, keeps the same with MaskGit. Moreover, they share a similary model architecture in terms of image generation. Therefore, we tend to believe that our Mask-GRPO can also work well in other MGMs like MaskGIT.
>
> Furthermore, we are trying to apply Mask-GRPO on Meissonic. However, due to limited time and computational resource, we have not finished it yet. We promise we will keep updating it. Thanks for your understanding.
>
> Thanks for your question.
>
> # Q4: Sample Efficiency and Compute Costs
> > The method introduces strategies like iteration reduction $\cdots$ how much performance was sacrificed?
>
> Thanks for your valuable questions. To be brief, **the unmasking reduction strategy achieves convergence using only 20% of the samples, resulting in a 25% performance improvement on GenEval while reducing training costs by 60%–80%**; while sample filtering does not aim to reduce the training cost, instead it **stablizes the training and gains a improved performance, as shown in Figure 4 (in the paper)**
>
> We begin with a brief recap of the definition of our unmasking reduction strategy: it reduces the total number of unmasking iterations during training (in our experiment, from $T$ = 50 to $T$ = 10), while maintaining the full unmasking schedule during evaluation. Importantly, in GRPO-based training (our method), one sample does not correspond to a single prompt or image prompt or one image, but rather to a single unmasking iteration, since the loss is computed iteratively at each step (see Equation (10)). Therefore, by applying the unmasking reduction strategy, we reduce both the number of unmasking iterations and the sample count to just 20%. Despite using only 20% of the samples, this strategy delivers a 25% performance gain onGenEval. Moreover, it leads to a 60%~80% reduction in training cost. To be more specific, it reduces the training hours from 230 A100 GPU hours to 80 hours in our experiments.
>
> For sample filtering, we discard low-variance sample groups to better highlight relative advantages $A$. This **enhances learning signal and stability**, ultimately leading to improved performance.
>
> Thanks for your question.
>
> # Q5: Theoretical Justification or Guarantees
> > Although the GRPO framework $\cdots$ expected to perform better than PPO or DPO in MGMs?
>
> Thanks for your insightful questions. The theoretical advantages of GRPO over PPO primarily stem from reduced bias and improved training stability, as discussed in [3,4]. These properties are particularly valuable in the masked generative modeling (MGM) setting, where multi-step iterative generation introduces long-term credit assignment challenges.
>
> We appreciate your interest in the theoretical underpinnings and plan to explore more rigorous justifications in future work.
>
> # Reference
> [1] Xu, Jiazheng, et al. "Imagereward: Learning and evaluating human preferences for text-to-image generation." Advances in Neural Information Processing Systems 36 (2023): 15903-15935.
>
> [2] Wang, Yibin, et al. "Unified reward model for multimodal understanding and generation." arXiv preprint arXiv:2503.05236 (2025).
>
> [3] Yue, Yu, et al. "Vapo: Efficient and reliable reinforcement learning for advanced reasoning tasks." arXiv preprint arXiv:2504.05118 (2025).
>
> [4] Yu, Qiying, et al. "Dapo: An open-source llm reinforcement learning system at scale." arXiv preprint arXiv:2503.14476 (2025).

---

> ### Author Response · Authors · 2025-08-02
> **Update on Meissonic**
>
> Dear Reviewer 5UPe,
>
> We have completed the experiment on Meissonic [1], and would like to share some updates related to the experiment as well as other aspects of our work.
>
> # Update on Q3
>
> We are pleased to report that we have applied Mask-GRPO to Meissonic [1] and evaluated it on GenEval benchmark. The results are summarized in the table below:
>
> **Table 9.** Mask-GRPO's GenEval Performance on Meissonic[1].
> |Model|Reward Model|Single Obj.|Two Obj.|Counting|Colors|Position|Color Attri.|Overall|
> |-|-|-|-|-|-|-|-|-|
> |Meissonic|-|**0.99**|0.66|0.42|0.86|0.10|0.22|0.54|
> |+ Mask-GRPO|CLIP|**0.99**|0.84|0.70|**0.89**|0.29|0.53|0.71|
> |+ Mask-GRPO|ImageReward[2]|**0.99**|**0.86**|**0.75**|**0.89**|0.33|0.51|**0.72**|
> |+ Mask-GRPO|UnifiedReward[3]|**0.99**|0.76|0.63|0.88|**0.34**|**0.54**|0.68|
>
> Note that we experimented with three different reward models, all of which **led to improvements over the baseline Meissonic [1], achieving up to a 33% increase in overall performance**. In all experiments, we used a learning rate of $3e-6$ and set the iteration steps to 64. These results further **demonstrate Mask-GRPO's generalizability across different MGM architectures**.
>
> # Update on Limitations
>
> Thank you for your thoughtful feedback. **We will incorporate these into the Limitations section**. For the misalignments with human judgment, we will integrate this with our response to Q2 (Robustness to Reward Models). For the potential reward hacking, we will elaborate a deeper discussion based on what we have discussed in l.292.
>
> Thank you again for your time and constructive review. We look forward to your further feedback!
>
> # Reference
> [1] Bai, Jinbin, et al. "Meissonic: Revitalizing masked generative transformers for efficient high-resolution text-to-image synthesis." The Thirteenth International Conference on Learning Representations. 2024.
>
> [2] Xu, Jiazheng, et al. "Imagereward: Learning and evaluating human preferences for text-to-image generation." Advances in Neural Information Processing Systems 36 (2023): 15903-15935.
>
> [3] Wang, Yibin, et al. "Unified reward model for multimodal understanding and generation." arXiv preprint arXiv:2503.05236 (2025).

---

> > ### Comment · Reviewer_5UPe · 2025-08-05
> >
> > I would like to thank the authors for their work and for answering all of my questions.

---

> > > ### Author Response · Authors · 2025-08-05
> > > **Gratitude and a Kind Request**
> > >
> > > Dear Reviewer 5UPe,
> > >
> > > Thank you once again for your thoughtful review and for acknowledging our responses. **Your feedback and engagement throughout the review and discussion process have certainly improved our paper**, and we will be sure to incorporate these changes and clarifications into the final version.
> > >
> > > If there are any remaining concerns, **we would be more than happy to address them**. If you feel that your concerns have been resolved, we would sincerely appreciate it if you could **consider updating your score to reflect a positive assessment of our submission**.
> > >
> > > Thank you again for your time and consideration. **Your support is a great encouragement to us**.

---

> > > > ### Comment · Reviewer_5UPe · 2025-08-08
> > > >
> > > > I appreciate the authors’ thorough and detailed rebuttal as well as the new experimental results, especially regarding the clarification of the transition probability formulations, the evaluation with multiple reward models, and the extension to the  MGM. These additions significantly strengthen the paper’s contributions and its claims of generalizability and robustness.
> > > >
> > > >
> > > > The transition probability definitions, while theoretically sound, remain somewhat complex and might benefit from additional visual aids or simplified explanations in future revisions to aid reader comprehension.
> > > >
> > > >
> > > > The discussion of potential limitations, such as biases inherited from reward models like CLIP or risks of reward hacking during long training, could be more explicitly elaborated to acknowledge societal and ethical considerations.

---

> > > > > ### Author Response · Authors · 2025-08-08
> > > > > **Feedback on Transition Probability Explanation and Limitation (Part 1/2)**
> > > > >
> > > > > Dear Reviewer 5UPe,
> > > > >
> > > > > Thank you very much for your time and thoughtful feedback. We are pleased that you acknowledged our previous rebuttal and new experimental results, and we sincerely appreciate your additional constructive suggestions. Please find our detailed responses below.
> > > > >
> > > > > # Transition Probability Explanation in Appendix
> > > > > > The transition probability definitions, while theoretically sound, remain somewhat complex and might benefit from additional visual aids or simplified explanations in future revisions to aid reader comprehension.
> > > > >
> > > > > Thank you for this valuable suggestion. We have **added a more in-depth explanation of our transition probability definitions in the Appendix**, in conjunction with Figure 2 (in the main text). Due to the regulation, we are not allowed to upload an updated manuscript in Openreview, but we we will incorporate these revisions in the next version. Here is the detail.
> > > > >
> > > > > - Added Start:
> > > > >
> > > > >  We provide a more in-depth explanation of our transition probability definitions ($p_{\theta 1}$ in Equation (14) and $p_{\theta 2}$ in Equation (15)) in this section. In brief, Equation (14) computes the sum of probabilities over all possible sampling paths that result in the same transition. The first term of the right-hand side represents the product of probabilities for newly unmasked tokens (patches), while the second term corresponds to the product of probabilities for newly remasked tokens. For Equation (15), it only computes the first term, focusing solely on the newly unmasked tokens and ignoring others.
> > > > >
> > > > > Let us explain these with Figure 2. In Figure 2, we compute the transition probability $p_{\theta 1}$ for iteration $t = 0$, which is $p_\theta (s_{1} | s_0, a_0)$ in Equation (14). For simplicity, assume the image consists of only 4 patches (index from 1 to 4), and let $i$ denote the patch index. Also assume that the visual codebook size is 4, with token ids from 1 to 4, denoted by $k$. We define $p_t^i$ as the softmax distribution at index $i$ at iteration $t$, and $(p_t^i)^k$ as the probability of sampling token id $k$ at index $i$. At $t = 0$, the initial state $s_0$ is fully masked, and only one patch needs to be unmasked.
> > > > >
> > > > > As shown in Figure 2, assume the softmax distributions are:
> > > > >
> > > > > - $p_0^1$ = $[0.1, 0.2, 0.6, 0.1]^T$
> > > > >
> > > > > - $p_0^2$ = $[0.5, 0.2, 0.1, 0.2]^T$
> > > > >
> > > > > - $p_0^3$ = $[0.3, 0.4, 0.2, 0.1]^T$
> > > > > - $p_0^4$ = $[0.15, 0.2, 0.15, 0.5]^T$.
> > > > >
> > > > > The key insight is that as long as the patch $i = 1$ samples token $k = 3$ with probability 0.6, regardless of what other patches sample, the resulting transition is always the same - first patch remains unmasked while the others are remasked. This is because when patch $i = 1$ samples token $k = 3$, its confidence score $cs_0^1$ = 0.6 is the highest among all $cs_0^i$, regardless of the other samples. According to Equation (5), this leads to the following Choose and Move (CaM) operations:
> > > > >
> > > > > - $CaM_0^1 = 1$
> > > > >
> > > > > - $CaM_0^2$ = $CaM_0^3$ = $CaM_0^4 = 0$
> > > > >
> > > > > In other words, the first patch is unmasked while the others are remasked.
> > > > >
> > > > > Returning to Equation (14), we compute the $p_\theta (s_{1} | s_0, a_0)$ based on Figure 2. Here we have:
> > > > >
> > > > > - $Y_0^{CaM}$ = {1}, which means the newly unmasked patch index with $CaM_0^i = 1$
> > > > >
> > > > > - $Y_0^M \backslash Y_0^{CaM}$ = {2, 3, 4}, which means the newly remasked patch index with $CaM_0^i = 0$
> > > > >
> > > > > Then the first term of Equation (14) is simply $cs_0^1$ = 0.6, representing the 'probability product of newly unmasked patches'. For the second term, we can rewrite it as $\prod_{i = 2, 3, 4} (\sum_{k} (p_0^i)^k)$, where k needs to be $(p_0^i)^k < min (cs_0) = cs_0^1$ = 0.6. However all $k = 1 , 2, 3, 4$ meet this requirment, no matter what $i$ is. Therefore, the second terms equals to $(0.5+ 0.2 + 0.1 + 0.1) \times (0.3 + 0.4 + 0.2 + 0.1) \times (0.15 + 0.2 + 0.15 + 0.5) = 1$, which is the 'probability product of newly remasked patches'. Putting it all together we have $p_{\theta 1}$ = $p_\theta (s_{1} | s_0, a_0) = 0.6 \times 1 = 0.6$, which is obvious illustrated in Figure 2. For $p_{\theta 2}$, it only includes the first term, giving $p_{\theta 2}$ = $cs_0^1$ = 0.6.
> > > > >
> > > > > - Added End
> > > > >
> > > > > We hope this detailed explanation and visual aid will significantly improve reader comprehension. Thank you again for the helpful suggestion.

---

> > ### Public Comment · ~Tobe_Daems1 · 2025-10-29
> >
> > Hi, wanna ask which dataset you used here for Meissonic? I found the base model is sensitive to some datasets.

---

> > > ### Public Comment · ~Yifu_Luo1 · 2025-10-30
> > >
> > > Hi, we used an internal dataset here. The reason was that we found Meissonic got moderate improvement using the dataset we used for show-o, so we attempted a higher quality dataset and obtained a better result.

---

> ### Author Response · Authors · 2025-08-08
> **Feedback on Transition Probability Explanation and Limitation (Part 2/2)**
>
> # Limitations
> > The discussion of potential limitations, such as biases inherited from reward models like CLIP or risks of reward hacking during long training, could be more explicitly elaborated to acknowledge societal and ethical considerations.
>
> Thank you for raising this important point. We have **expanded the discussion of limitations in the 'Conclusion and Futhur Work' Section**. Due to the regulation, we are not allowed to upload an updated manuscript in Openreview, but we we will incorporate these revisions in the next version. Here is the detail.
>
> - Before:
>
> While achieving superior results, Mask-GRPO’s potential is still not fully revealed. On the one hand, applying such methods on larger base models to explore possible scaling-up will be interesting work. On the other hand, Mask-GRPO’s potential on video generation is valuable, while text-to-video generation is always more complicated as it has to generate coherent outputs that are both contextually and temporally consistent.
>
> - After:
>
> While achieving superior results, Mask-GRPO’s potential is still not fully revealed. First, unlike RL post-training in LLMs, applying RL to T2I generation relies on effective reward models to evaluate generated images. In this work, we use CLIP as the reward model, which proves effective. However, like other reward models (e.g., ImageReward, UnifiedReward), CLIP is still narrow in scope and not accurate enough for further improvements in RL post-training, especially considering the societal biases inherited from the pre-training data. Developing more robust and fair reward models, or jointly training them during RL, may help mitigate these issues. Second, although our out-of-domain results (on GenEval and MSCOCO) suggests that Mask-GRPO successfully mitigates reward hacking, ensuring long-term resistance to such behavior remains an open challenge. Over extended training, models may learn to exploit weaknesses in the reward function, potentially leading to degraded or unsafe outputs. Incorporating human feedback or adversarial training may be necessary to guard against this risk. Third, applying Mask-GRPO to larger base models to explore possible scaling-up will be an interesting work, as RL post-training may benefit from improved model capacity. Finally, Mask-GRPO also holds potential for video generation, where the challenge lies in maintaining both temporal coherence and semantic consistency — a more complex objective than in still-image generation.
>
> We believe this revision **better acknowledges the societal and ethical considerations, especially regarding reward models and reward hacking**. Thank you for helping us improve this important section.
>
> # Conclusion
>
> **Thank you once again for your time, constructive feedback, and encouragement** throughout the review process. Your feedback and engagement has significantly improved the clarity and quality of our work. We hope our responses have addressed all your concerns, and we **welcome any further discussion or questions you may have**.
>
> If all your questions have been satisfactorily addressed, we would be **deeply grateful if you could consider increasing your score to reflect a more positive assessment**. Your support would be a tremendous encouragement to us.
>
> Thank you!

---

> ### Author Response · Authors · 2025-08-08
> **Sincerely Looking Forward to Your Feedback**
>
> Dear Reviewer 5UPe,
>
> We would like to sincerely thank you once again for your valuable feedback, insightful comments, and the time and effort you’ve dedicated to reviewing our work. Your constructive suggestions have been immensely helpful in improving our paper.
>
> As the discussion period is coming to a close in about 24 hours, we wanted to kindly check if our response has fully addressed your concerns. If there are any remaining questions or points for discussion, we would be more than happy to engage further.
>
> If our response has resolved your concerns, we would be truly grateful if you might consider adjusting your score. Your support would mean a great deal to us!

---

> ### Author Response · Authors · 2025-08-09
> **Kindly ask any remaining issues?**
>
> Dear Reviewer 5UPe,
>
> Thank you again for reading our rebuttal and for the valuable suggestions. **We noticed that you have submitted final score. We would like to kindly ask whether our rebuttal has resolved your concerns and whether there are any remaining issues you would like us to clarify**. We sincerely hope that you will consider both our rebuttal content and our original paper. We would appreciate any additional feedback.

---

### Official Review · Reviewer_Ydx6 · 2025-06-26

**Clarity:** 3
**Significance:** 3
**Originality:** 3
**Rating:** 4
**Confidence:** 3

**Summary:**

This paper introduces Mask-GRPO, a method to incorporate Group Relative Policy Optimization (GRPO)-based online reinforcement learning (RL) into masked generative models (MGMs) for text-to-image (T2I) generation. The authors redefine the transition probability in RL for MGMs, formulating the unmasking process as a multi-step decision-making problem. They propose two candidate definitions for transition probability and explore several enhancement strategies, including removing the Kullback-Leibler (KL) constraint, applying a reduction strategy, and filtering out low-quality samples. The method shows improvement compare with the base model (Show-o) on the GenEval benchmark and  MSCOCO-30K FID.

**Questions:**

1. Can you offer more details on equation (14)?
2. Can you provide more evaluations on T2I tasks other than Geneval?

**Ethical Concerns:**

["NO or VERY MINOR ethics concerns only"]

**Final Justification:**

Authors address my concerns, so I maintain my score.

**Limitations:**

Yes.

**Paper Formatting Concerns:**

No.

**Quality:**

3

**Strengths And Weaknesses:**

### Strengths
1. The experiment explores different probability calculation methods when treating the generation process of MGM as a Markov process, with a clear motivation.

2. According to the results reported in the paper, Mask-GRPO improves the performance of show-o on Geneval and COCO-30K FID.

3. The paper is well-structured, with clear explanations of the methodology.

### Weaknesses
1. The font size in the figures is too small—it should ideally match the font size of the main text.

2. Equation (14) appears to be the core part of the proposed method. However, the details of Equation (14) in the paper are not sufficiently clear. Could more details about Equation (14) be provided?

3. In addition to the evaluation on Geneval, could more evaluations on the latest text-to-image (t2i) benchmarks, such as DPG-Bench and WISE, be included? Geneval has actually been around for a long time, and evaluation on newer benchmarks would better demonstrate the method's effectiveness.

4. There are some spelling errors in the paper, such as "in-troduce" in line 79 and "settingβ=0" in line 212.

---

> ### Author Rebuttal · Authors · 2025-07-31
>
> Dear reviewer Ydx6,
>
> Thank you for your comprehensive review of our paper. We greatly appreciate your recognition of the clear motivation and strong empirical results of our method. Please find our detailed responses to your concerns below.
>
> # Q1: More Details on Equation (14)
> > Can you offer more details on equation (14)?
>
> Thank you for your question. We would like to offer a more in-depth explanation of Equation (14), in conjunction with Figure 2 (in our paper). In brief, Equation (14) **computes the sum of probabilities over all possible sampling paths that result in the same transition. The first term of the right-hand side represents the product of probabilities for newly unmasked patches (tokens), while the second term corresponds to the product of probabilities for newly remasked patches**.
>
> Let us first revisit the definition. In Figure 2, we compute the transition probability for iteration $t = 0$, which is $p_\theta (s_{1} | s_0, a_0)$ in Equation (14). For simplicity, assume the image consists of only 4 patches (index from 1 to 4), and let $i$ denote the patch index. Also assume that the visual codebook size is 4, with token ids from 1 to 4, denoted by $k$. We define $p_t^i$ as the softmax distribution at index $i$ at iteration $t$, and $(p_t^i)^k$ as the probability of sampling token id $k$ at index $i$. At $t = 0$, the initial state $s_0$ is fully masked, and only one patch needs to be unmasked.
>
> As shown in Figure 4, assume the softmax distributions are:
>
> - $p_0^1$ = $[0.1, 0.2, 0.6, 0.1]^T$
>
> - $p_0^2$ = $[0.5, 0.2, 0.1, 0.2]^T$
>
> - $p_0^3$ = $[0.3, 0.4, 0.2, 0.1]^T$
> - $p_0^4$ = $[0.15, 0.2, 0.15, 0.5]^T$.
>
> The key insight is that **as long as the patch $i = 1$ samples token $k = 3$ with probability 0.6, regardless of what other patches sample, the resulting transition is always the same - first patch remains unmasked while the others are remasked**. This is because when patch $i = 1$ samples token $k = 3$, its confidence score $cs_0^1$ = 0.6 is the highest among all $cs_0^i$, regardless of the other samples. According to Equation (5), this leads to the following Choose and Move (CaM) operations:
>
> - $CaM_0^1 = 1$
>
> - $CaM_0^2$ = $CaM_0^3$ = $CaM_0^4 = 0$
>
> In other words, the first patch is unmasked while the others are remasked.
>
> Returning to Equation (14), we compute the $p_\theta (s_{1} | s_0, a_0)$ based on Figure 2. Here we have:
>
> - $Y_0^{CaM}$ = {1}, which means the newly unmasked patch index with $CaM_0^i = 1$
>
> - $Y_0^M \backslash Y_0^{CaM}$ = {2, 3, 4}, which means the newly remasked patch index with $CaM_0^i = 0$
>
> Then the first term of Equation (14) is simply $cs_0^1$ = 0.6, representing the 'probability product of newly unmasked patches'. For the second term, we can rewrite it as $\prod_{i = 2, 3, 4} (\sum_{k} (p_0^i)^k)$, where k needs to be $(p_0^i)^k < min (cs_0) = cs_0^1$ = 0.6. However all $k = 1 , 2, 3, 4$ meet this requirment, no matter what $i$ is. Therefore, the second terms equals to $(0.5+ 0.2 + 0.1 + 0.1) \times (0.3 + 0.4 + 0.2 + 0.1) \times (0.15 + 0.2 + 0.15 + 0.5) = 1$, which is the 'probability product of newly remasked patches'. Putting it all together we have $p_\theta (s_{1} | s_0, a_0) = 0.6 \times 1 = 0.6$, which is obvious illustrated in Figure 2.
>
> To summrize, Equation (14) accurately computes the real transition probability, which AR-style (see Equation (13)) cannot capture. It explicitly distinguishes between newly unmasked and newly remasked patches and calculates their probabilities accordingly.
>
> We sincerely appreciate your question and apologize for the lack of detail of Equation (14). These clarifications will be incorporated into the revised version.
>
> # Q2: Other Benchmarks
> > Can you provide more evaluations on T2I tasks other than Geneval?
>
> Thank you for this valuable suggestion. We have conducted additional experiments on WISE[1], The results are as follows:
>
> **Table 7.** Model Comparison on WISE[1].
> |Model|Params|Culture|Time|Space|Biology|Physics|Chemistry.|Overall|
> |-|-|-|-|-|-|-|-|-|
> |Show-o|1.3B|0.28|0.40|0.48|0.30|0.46|**0.30**|0.35|
> |**Show-o + Mask-GRPO**|1.3B|**0.53**|**0.48**|0.46|0.41|0.35|0.21|**0.45**|
> |vlla-u-7b-256|7.0B|0.26|0.33|0.37|0.35|0.39|0.23|0.31|
> |Orthus-7B-base|7.0B|0.07|0.10|0.12|0.15|0.15|0.10|0.10|
> |Orthus-7B-instruct|7.0B|0.23|0.31|0.38|0.28|0.31|0.10|0.10|
> |Janus-Pro-1B|1.0B|0.20|0.28|0.45|0.24|0.32|0.16|0.26|
> | Janus-Pro-7B|7.0B| 0.30     | 0.37 | 0.49  | 0.36    | 0.42    | 0.26      | 0.35    |
> | JanusFlow-1.3B |1.3B | 0.13     | 0.26 | 0.28  | 0.20    | 0.19    | 0.11      | 0.18    |
> | Harmon-1.5B|1.5B  | 0.38     | **0.48** | **0.52**  | 0.37    | 0.44    | 0.29      | 0.41    |
> | SD-v1-5  |1.0B  | 0.34     | 0.35 | 0.32  | 0.28    | 0.29    | 0.21      | 0.32    |
> | SD-2-1   |1.5B     | 0.30     | 0.38 | 0.35  | 0.33    | 0.34    | 0.21      | 0.32    |
> | SD-XL-base-0.9 |N/A     | 0.43     | **0.48** | 0.47  | **0.44**    | 0.45    | 0.27      | 0.43    |
> | SD-3-medium |N/A    | 0.42     | 0.44 | 0.48  | 0.39    | **0.47**    | 0.29      | 0.42    |
> | FLUX.1-schnell|12B      | 0.39     | 0.44 | 0.50  | 0.31    | 0.44    | 0.26      | 0.40    |
>
> As shown, Mask-GRPO achieves substantial gains in WISE, with up to 29% improvement, demonstrating the generalizability of our method beyond GenEval.
>
> However, due to time and computational constraints, we have not yet completed the evaluation on DPG-Bench. We will update the results in the second phase of the rebuttal. Thank you for your understanding.
>
> We will include these new benchmarks in the revised version of our paper. Thank you again for the helpful suggestion.
>
> # W1: Font size
> > The font size in the figures $\cdots$ match the font size of the main text.
>
> Thank you. We will revise all figures to ensure the font sizes are consistent with the main text in the updated manuscript.
>
> # W4: Spelling Errors
> > There are some spelling errors $\cdots$ in line 212.
>
> Thank you for pointing this out. We will thoroughly proofread the manuscript and correct all typographical and grammatical issues. We sincerely apologize for any inconvenience this may have caused in your reading experience.
>
> # References
> [1] Niu, Yuwei, et al. "Wise: A world knowledge-informed semantic evaluation for text-to-image generation." arXiv preprint arXiv:2503.07265 (2025).

---

> > ### Comment · Reviewer_Ydx6 · 2025-08-05
> >
> > Thanks for your response. I'll maintain my score.

---

> > > ### Author Response · Authors · 2025-08-05
> > > **Thanks!**
> > >
> > > Dear Reviewer Ydx6,
> > >
> > > Thank you once again for your response, your continued engagement, and your commitment to helping improve our paper. We truly appreciate the time and effort you have dedicated to reviewing our work.

---

> ### Author Response · Authors · 2025-08-01
> **Update on Q2 (DPG-Bench)**
>
> # Update on Q2
> Dear reviewer Ydx6,
>
> We are pleased to report that we have completed the evaluation of Mask-GRPO on DPG-Bench[2], and the results are summarized in the table below:
>
> **Table 8.** Model Comparison on DPG-Bench[2].
> |Model|Params|Entity|Other|Global|Attribute|Relation| **Overall** |
> |-|-|-|-|-|-|-|-|
> |Show-o|1.3B|72.97|73.93|73.76|73.59|72.85|67.48|
> |**Show-o + Mask-GRPO**|1.3B|**90.63**|**89.87**|90.89|**90.03**|**91.85**|**85.63**|
> |PixArt-alpha|0.6B|79.32|76.96|74.97|78.60|82.57|71.11|
> |SDv2.1|0.9B|78.13|80.66|77.67|74.91|80.72|68.09|
> |SDXL|2.6B|82.43|80.41|83.27|80.91|86.76|74.65|
> |SD3.5-M|2B|-|-|-|-|-|84.08|
> |DALL-E 3|-|89.61|89.83|90.97|88.39|90.58|83.50|
> |LlamaGen|0.8B|-|-|-|-|-|65.16|
> |Janus|1.5B|-|-|82.33|-|85.46|79.68|
> |SimpleAR-1.5B|1.5B|-|-|87.97|-|88.33|81.97|
> |Infinity|2B|-|-|**93.11**|-|90.76|83.46|
> |Emu3|8.5B|-|-|-|-|-|81.60|
>
> As shown in Table 8, integrating Mask-GRPO with the base Show-o model leads to a significant performance gain, improving the overall score on DPG-Bench from 67.48 to 85.63 — **a 27% increase**. Notably, our method outperforms many other state-of-the-art approaches, demonstrating its strong and consistent effectiveness. We hope this additional evidence clarifies the strength of our approach.
>
> Thank you once again for your time and thoughtul review. We look forward to hearing from you!

---

### Official Review · Reviewer_Vae1 · 2025-06-29

**Clarity:** 2
**Significance:** 3
**Originality:** 3
**Rating:** 5
**Confidence:** 3

**Summary:**

This paper proposes to apply Group Relative Policy Optimization (GRPO) to text-to-image Masked Generative Models (MGMs). Specifically, the authors frame MGM inference as a Markov decision process, and explore different options to implement the transition probability between decoding steps.

In addition, they revisit various reinforcement learning training practices and optimizations in this new masked generative modeling context.

For experiments, they start from a Show-o base model, use CLIP as the reward function, train on LAION text prompts, and evaluate performance on GenEval and MSCOCO-30K FID.

**Questions:**

Questions are moslty related to my concerns presented here-above:
- Are there any regularities in the types of prompts that tend to fail the variance threshold? If a resampled group still fails to pass, what happens? Is it repeatedly resampled?
- What mechanism do the authors believe allows Mask-GRPO to improve on GenEval metrics, and counting and attribute binding tasks, while only relying on CLIP as the reward model?
- Could the authors provide the text prompts corresponding to the samples shown in Figure 5? This would help readers better interpret the qualitative results, especially w.r.t. the comment about diversity l.299.

**Ethical Concerns:**

["NO or VERY MINOR ethics concerns only"]

**Final Justification:**

I find the premise of the paper very interessing, and the discussions about GRPO could elevate CLIP as a judge particularly enlightning.
Moreover, my comments have been very thoroughly adressed.
I hope the paper is accepted.

**Limitations:**

yes

**Paper Formatting Concerns:**

No major formatting issues.

**Quality:**

2

**Strengths And Weaknesses:**

Strength
- Applying GRPO to masked generative models is, to my knowledge, a novel proof of concept. I would add that, unlike autoregressive and diffusion models -- where the training objective and inference process are naturally aligned -- masked generative models suffer from a training–inference mismatch. Reinforcement learning provides a principled way to bridge that gap, making this contribution particularly interesting.
- The authors experiment with different formulations for the transition probability between decoding steps, and systematically reassess the impact of multiple RL training stabilizations and optimizations in this setting.
- Mask-GRPO achieves very competitive improvements over the base Show-o model on standard text-to-image metrics.

Weaknesses
- I find the paper not very clearly presented, with some formulations that are confusing or imprecise. For instance, Equation 13 is described as “the product of every [masked] token probability” (line 165), but also as “the joint probability of all newly predicted tokens” (line 37), which seems contradictory. The second phrasing is potentially misleading, as it could be interpreted to mean the method is modeling the joint distribution of token values (as is the goal in generative modeling, e.g., autoregressive or diffusion models), whereas, as far as I understand, the policy is actually modeling the probability of selecting tokens for unmasking. Clarifying this distinction would avoid confusion.

- The paper claims that low reward variance within a group motivates the sample filtering approach. While the filtering is ablated and shown to be effective, it would be useful to back this initial claim, maybe with qualitative examples, for example, showing sets of images of different variance.

- Related to sample filtering, it is unclear whether resampled groups are repeatedly evaluated and rejected until they pass the variance threshold. If so, are there certain bad prompts that never pass and get stuck?

- Using CLIP as a reward model could inherently limit the improvements Mask-GRPO can achieve, given CLIP’s well-established weaknesses such as counting and attribute binding. In fact, GenEval was explicitly designed to address the shortcomings of the CLIP score. As such, it seems surprising that the proposed method improves performance across the board on GenEval, and even shows gains in counting and attribute binding (as suggested by Figure 3). This apparent contradiction should be discussed in more detail.

- There seems to be quite a few typos. Some I noted are
l.296. presented, l.235  seems to be lacking a figure reference, l.229 includes an unnecessary dot inside a sentence

---

> ### Author Rebuttal · Authors · 2025-07-31
>
> Dear Reviewer Vae1,
>
> Thank you very much for your comprehensive and detailed review of our paper, as well as for recognizing the novelty and effectiveness of our work. Please find our point-by-point responses below.
>
> # Q1: Sample Filter
> > Are there any regularities in the types of prompts $\cdots$ Is it repeatedly resampled?
>
> Thank you for your detailed question. Yes, if a sample group is rejected, it will **repeatdly resample itself until it either passes the varience threshold or reaches the maximum number of attempts** (set to 3 in our experiments to balance computation costs).
>
> In our experiments, we observed that **certain prompts consistently fail to meet the variance threshold within the allowed number of attempts**. Below are several representative examples:
>
> - '6ad74e6708fc4 Dresses For 14 Year Girls Online Shopping | Dresses For 14 Year ...'
> - 'Halloween-Images-For-Kids-Clip-Art.png'
> - 'Tags: K-Pop, Kim Chung-ha, Serious, Make Up, Lemon, Blue Background, Nail Polish, Fruits, Yellow Outfit, Yellow Dress, Blooming Blue'
> - '6947 Cooper Point Rd NW, Olympia, WA 98502 (#1486235) :: Platinum Real Estate Partners'
> - 'How to beat travel stress on family holidays'
> - 'Round Trivet - #25 - Tree of Life (3 colour option)'
> - 'Beautiful 1970s boho lace maxi dress'
> - 'How to Determine Your Body Type | Jalisa's Fashion Files'
> - 'Palm tree clip art'
> - 'Watercolor Holiday Characters - image 4 - student project'
> - '2018 Transit 150 Low Roof,  Empty Cargo Van #T80223 - photo 5'
>
> We found that these prompts share some common characteristics:
>
> - **Non-Descriptive or Unstructured Prompts**: Many appear to be taken directly from web page titles or article headers (e.g., ’6ad74e6708fc4 Dresses For 14 Year Girls Online Shopping | Dresses For 14 Year ...‘), often including irrelevant or noisy content, and lacking clear visual instruction.
>
> - **Lack of Visual Specificity and High Ambiguity**: Some prompts are abstract or stylistically vague. They do not clearly specify the visual information, which causes high ambiguity. For instance, 'How to beat travel stress on family holidays' is conceptually broad with no clear visual target; 'Palm tree clip art' suggests a style but does not describe the image clearly; and 'Halloween-Images-For-Kids-Clip-Art.png' leaves many details undefined (e.g., number of characters, composition, setting).
>
> To summarize, these 'hard-pass' prompts lack of specific visual guidance, which leads to high ambiguity and frequent rejection by our variance-based filter.
>
> Thank you again for the question. We hope we clarify the details well.
>
> # Q2: CLIP Model
> > What mechanism do  $\cdots$ only relying on CLIP as the reward model?
>
> Thank you for this insightful question. It is well known that the CLIP model has limited ability in counting and attribute binding. However, we believe the success of our method stems from the design of GRPO, where **we do not rely on absolute correctness, but rather on relative correctness within a sample group.**
>
> In GRPO-based methods (ours), we do not use raw rewards from the reward model directly. Instead, we transform them to group-level normalized advantage $A_t^j$ (see Equation (12)), which is effectively **comparing the relative correctness among samples within a group** [1].  Therefore, even if CLIP model cannot provide an accurate 'absolute correctness' for counting or attribute correctness, as long as it **capture the relative trend**, it can still drive the model to improve.
>
> To verify this, we conducted an additional experiment. We randomly selected 10 GenEval-counting-style prompts (e.g. 'a photo of $N$ items') and generated 10 correct images accordingly. We then modified each prompt to 'a photo of $\frac{N}{2}$ items' and 'a photo of $2N$ items', while keeping the originally generated (correct) images. We computed the CLIP scores for each prompt-image pair using our reward model. The results are as follows:
>
> **Table 5.** Additional Experiments on Relative Correctness
> | Original Prompts | $\frac{N}{2}$ | $N$ | $2N$ |
> | ----- | ------ |------ |------ |
> | 'a photo of six clocks' | 32.5298 | 35.8555 | 33.2857 |
> | 'a photo of four handbags' | 29.7800 | 33.0426 | 32.4762 |
> | 'a photo of two backpacks' | 28.4991 | 34.1256 | 29.7021 |
> | 'a photo of two frisbees' | 32.1401 | 36.8218 | 29.6494 |
> | 'a photo of four toothbrushs' | 33.8095 | 38.2953 | 34.8197 |
> | 'a photo of four vases' | 28.2428 | 37.1507 | 29.9521 |
> | 'a photo of two computer keyboards' | 32.3883 | 37.2445 | 31.1868 |
> | 'a photo of four baseball gloves' | 29.5035 | 33.6917 | 31.3151 |
> | 'a photo of two beds' | 33.5974 | 38.9766 | 31.7446 |
> | 'a photo of four giraffes' | 29.2052 | 36.3132 | 32.4882 |
> | **Overall** | **30.9696** | **35.1873** | **31.6620** |
>
> As shown, CLIP consistently assigns higher scores to prompts matching the correct count, demonstrating that it can indeed capture 'relative correctness'. This justifies the use of CLIP within our GRPO framework and explains why Mask-GRPO can improve performance in counting and attribute-binding tasks.
>
> We appreciate your valuable question and hope this additional experiment provides further clarity.
>
> # Q3: Text Prompts of Figure 5
> > Could the authors provide the text prompts $\cdots$ the comment about diversity l.299.
>
> Thank you for your careful suggestion. Below are the text prompts corresponding to each image in Figure 5 (each line, from left to right):
>
> Line 1:
>
> - Create an image of a cat as a gardener, wearing a straw hat, gardening gloves, and surrounded by colorful flowers
>
> - Anime illustration of Princess Mononoke from Studio Ghibli, by artgerm, stunning artwork
>
> - Futuristic cyberpunk city at night, neon lights, high-tech car, vibrant colors, cinematic lighting, highly detailed, sci-fi atmosphere, 8k resolution, unreal engine
>
> - A whimsical pink cloud-shaped building with minimalist windows and doors, floating above a vibrant blue sky with cotton-like clouds, Studio Ghibli-style animation movie texture
>
> Line 2:
>
> - A white puppy sitting playfully in autumn leaves, surrounded by fallen red apples, soft natural lighting
>
> - Heroic elf warrior, golden glowing background, detailed fantasy armor, cinematic lighting, epic fantasy art, high detail
>
> - Vibrant city skyline during sunset, modern skyscrapers, colorful abstract style, warm gradient sky, digital art, urban landscape, vivid colors
>
> - Mystical forest with glowing mushrooms and a babbling brook
>
> Line 3:
>
> - Futuristic metallic humanoid robot, highly detailed face, sci-fi background, cinematic lighting, dystopian cityscape, 4K resolution
>
> - A cute duck wearing a chef uniform covered in cookie batter, unreal engine render 8k
>
> - A realistic Venusaur animal among the trees, forest lake, moss, cold weather, dark teal and amber, Sony A7 IV
>
> - 4d photographic image of full body image of a cute little chibi boy realistic, vivid colors octane render trending on artstation, artistic photography, photorealistic concept art, soft natural volumetric cinematic perfect light, UHD no background
>
> We sincerely apologize for the omission and will include these prompts in the revised version.
>
> # W1: Unclear Presentation
> > I find the paper not very clearly presented $\cdots$ Clarifying this distinction would avoid confusion.
>
> Thanks for your suggestion. We agree that the phrasing in l.37 may be misleading. We will revise this sentence to clarify the distinction and avoid potential confusion. We sincerely apologize for any inconvenience caused.
>
> # W2: Qualitative Presentation
> > The paper claims that low reward variance $\cdots$ showing sets of images of different variance.
>
> Thank you for this valuable suggestion. We agree that visualizing image sets with different reward variances would better convey our ideas - particularly in combination with the discussion in Q2. We will include such examples in the revised version.
>
> # W5: Typos
> > There seems to be quite a few typos $\cdots$ inside a sentence.
>
> Thank you for pointing this out. We will thoroughly proofread the manuscript and correct all typographical and grammatical errors in the revision. We sincerely apologize for any disruption this may have caused to your reading experience.
>
> # References
> [1] Guo, Daya, et al. "Deepseek-r1: Incentivizing reasoning capability in llms via reinforcement learning." arXiv preprint arXiv:2501.12948 (2025).

---

> ### Comment · Reviewer_Vae1 · 2025-08-05
>
> I thank the authors for providing the requested details. I find them very insightful and hope they can be included in the paper or in the appendix.
>
> The CLIP model analysis is particularly nice, although N/2 and 2N might be a bit dramatic.
> A larger experimental campaign, possibly in future work, could verify if the preference provided by CLIP is statistically meaningful for smaller changes in numbers of objects.
>
> I would recommend the authors to provide more details about how they intend to clarify the wording, so that we can have more confidence that in will indeed be fixed.
>
> In any case, the rebuttal gives food for thoughts and I will reconsider the paper in light of these new elements.

---

> ### Author Response · Authors · 2025-08-06
> **Refinements on Wording, Typo Corrections, and CLIP Experiments (Part 1/2)**
>
> Dear Reviewer Vae1,
>
> Thank you very much for your time and effort in reviewing our paper. We are glad that you found our rebuttal useful, and we sincerely appreciate your further suggestions. Please find our detailed responses regarding the **wording**, **typos**, and **CLIP experiments** below.
>
> # Wording
> > I would recommend the authors to provide more details about how they intend to clarify the wording $\cdots$ indeed be fixed.
>
> Thank you for your valuable suggestion. We realized that much of the confusion in our presentation **stemmed from the ambiguous use of the term 'newly predicted tokens'** (e.g., l.21). To improve clarity, we have revised our manuscript as follows. Due to the regulation, we are not allowed to upload an updated manuscript in Openreview, but we we will incorporate these revisions in the next version.
>
> - l.21-l.24
>
> Before: they predict all masked tokens simultaneously in parallel at each iteration, defined as newly predicted tokens, but only keep the most confident ones, defined as newly unmasked tokens. The rest of the newly predicted tokens will be re-masked for the next iteration, which we define as newly re-masked tokens.
>
> After: they predict all masked tokens simultaneously in parallel at each iteration, but only keep the most confident ones, defined as newly unmasked tokens. The rest will be remasked for the next iteration, which we define as newly remasked tokens.
>
> - l.37-l.39
>
> Before: Since MGMs predict all unmasked tokens in parallel, a naive solution is to use the joint probability of all newly predicted tokens at each iteration as the transition probability $\cdots$
>
> After: Since MGMs predict all masked tokens in parallel, a naive solution is to use the product of their probabilities at each iteration as the transition probability $\cdots$
>
> - l.47-l.51
>
> Before: Building on this, we propose two candidate definitions: (1) the joint probability over all newly unmasked tokens at each iteration. (2) the joint probability over both newly unmasked and remasked tokens $\cdots$
>
> After: Building on this, we propose two candidate definitions: (1) the product of probability over all newly unmasked tokens at each iteration. (2) the product of probability over both newly unmasked and remasked tokens $\cdots$
>
> - l.104-l.106
>
> Before: At each iteration t, the model predicts all tokens in $Y^M_t$ simultaneously, which we defined as newly predicted tokens in Section 1, and moves a subset of them, $\cdots$
>
> After: At each iteration t, the model predicts all masked tokens in $Y^M_t$ simultaneously and moves a subset of them, $\cdots$
>
> - l.173-l.174
>
> Before: The newly predicted tokens (those with $CaM_t^i = 1$ in Equation (5)), play the most critical role $\cdots$
>
> After: The newly unmasked tokens (those with $CaM_t^i = 1$ in Equation (5)), play the most critical role $\cdots$
>
> - l.190-l.192
>
> Before: The sampling results for the newly predicted tokens $Y_t^{CaM}$ remain the same. The remaining tokens in $Y_t^M \backslash Y_t^{CaM} $, which will be remasked, have lower confidence scores $cs_t^i$ than any of newly predicted tokens in $Y_t^{CaM}$.
>
> After: The sampling for the newly unmasked tokens $Y_t^{CaM}$ remain the same. The remaining tokens in $Y_t^M \backslash Y_t^{CaM} $, which will be remasked, have lower confidence scores $cs_t^i$ than any of newly unmasked tokens in $Y_t^{CaM}$.
>
> - l.200-l.201
>
> Before: $\cdots$ focusing solely on the newly predicted tokens $Y_t^{CaM}$ and ignoring others.
>
> After: $\cdots$ focusing solely on the newly unmasked tokens $Y_t^{CaM}$ and ignoring others.
>
> We hope that these wording revisions help clarify our presentation by **removing the ambiguity surrounding the term 'newly predicted tokens'.**
>
> Once again, thank you for pointing this out.

---

> ### Author Response · Authors · 2025-08-06
> **Refinements on Wording, Typo Corrections, and CLIP Experiments (Part 2/2)**
>
> # Typos
>
> We sincerely apologize for the inconvenience caused by the typos in our submission. After carefully reviewing the manuscript, we have corrected the following issues. Due to the regulation, we are not allowed to upload an updated manuscript in Openreview, but we we will incorporate these revisions in the next version.
>
> - l.79
>
> Before: $\cdots$ first to in-troduce $\cdots$
>
> After: $\cdots$ first to introduce $\cdots$
>
> - l.196
>
> Before: $\cdots$  sampling the $k$-th at position $i$ at iteration $t$
>
> After: $\cdots$  sampling the $k$-th token at position $i$ at iteration $t$
>
> - l.229
>
> Before: $\cdots$ during training, e.g., from $T=50$ to $T=10$. while keeping $\cdots$
>
> After: $\cdots$ during training (e.g., from $T=50$ to $T=10$), while keeping $\cdots$
>
> - l.296
>
> Before: $\cdots$ are presneted in $\cdots$
>
> After: $\cdots$ are presented in $\cdots$
>
> - l.212
>
> Before: $\cdots$ effectively setting$\beta$=0 $\cdots$
>
> After: $\cdots$ effectively setting $\beta$=0 $\cdots$
>
> - l.235 : Added the figure for reward instability.
>
> - l.522 : Corrected reference formatting.
>
> - Figure 1 : Adjusted figure size and font size to match the main text.
>
> - Figure 4: Converted the figure content into a table, as detailed in our response to Reviewer QEdg (Q4).
>
> We once again apologize for these errors and greatly appreciate your attention to detail.
>
> # CLIP Experiment
> >  A larger experimental campaign, possibly in future work $\cdots$ in numbers of objects.
>
> Thank you for acknowledging our additional CLIP experiments and for your insightful suggestion. We will incorporate these into our paper.
>
> We are indeed exploring this line of work about reward models. Unlike RL post-training in LLMs, applying RL to text-to-image generation requires effective reward models for evaluating generated images. However, current options (e.g., CLIP, HPS, UnifiedReward $\cdots$) are either too narrow in scope or insufficiently accurate. Therefore, currently we are focusing on **jointly training the reward model during RL post-training**, rather than treating it as a frozen evaluator. We are excited to further explore this promising avenue and hope to report meaningful progress in future work.
>
> # Conclusion
>
> Once again, **thank you sincerely for your time, constructive feedback, and encouragement** throughout the review process. Your suggestions have been truly insightful and have helped us improve our work significantly. We hope that our clarifications have addressed your concerns. **We would be very grateful if you would consider increasing your support for our paper — your support would be a great motivation for our continued research**.
>
> Thank you!

---

> ### Author Response · Authors · 2025-08-08
> **Sincerely Looking Forward to Your Feedback**
>
> Dear Reviewer Vae1,
>
> We would like to sincerely thank you once again for your valuable feedback, insightful comments, and the time and effort you’ve dedicated to reviewing our work. Your constructive suggestions have been immensely helpful in improving our paper.
>
> As the discussion period is coming to a close in about 24 hours, we wanted to kindly check if our response has fully addressed your concerns. If there are any remaining questions or points for discussion, we would be more than happy to engage further.
>
> If our response has resolved your concerns, we would be truly grateful if you might consider increasing your support. Of course, we fully respect your decision either way and deeply appreciate your thoughtful review. Your support would mean a great deal to us!

---

> > ### Comment · Reviewer_Vae1 · 2025-08-09
> >
> > Yes, thank you for your thorough answers. I have no more comments and will re-assess.

---

> > > ### Author Response · Authors · 2025-08-09
> > > **Thanks!**
> > >
> > > Dear Reviewer Vae1,
> > >
> > > Thank you so much! Your response, your continued engagement, and your commitment have truly illuminated the path of our research. We deeply appreciate the time, expertise, and thoughtful consideration you have dedicated to reviewing our work. Your encouragement not only inspires us to strive for higher standards but also reminds us of the value of sincere dedication in scholarly pursuits. We will carry forward this spirit of genuine care and commitment in our future research, hoping to pass on the same support and inspiration to others in our field.
> > >
> > > Thank you!

---

### Official Review · Reviewer_QEdg · 2025-07-02

**Clarity:** 3
**Significance:** 3
**Originality:** 3
**Rating:** 4
**Confidence:** 4

**Summary:**

This work introduces Mask-GRPO, a reinforcement learning (RL) method specifically designed for masked generative models in text-to-image (T2I) generation. The proposed method applies Group Relative Policy Optimization (GRPO) in a novel way by framing the image unmasking process as a multi-step decision-making problem, incorporating a new definition of transition probability. The model's performance is further enhanced through several key strategies, including the removal of the Kullback–Leibler constraint, a reduction strategy, and the filtering of low-quality samples.

**Questions:**

See Weakness.

**Ethical Concerns:**

["NO or VERY MINOR ethics concerns only"]

**Final Justification:**

I thank the authors for their efforts on the manuscript and rebuttal. Before the rebuttal, I raised concerns about the statistical error, category discrepency, and \etc. The authors have addressed these concerns well in their response. Therefore, I maintain the borderline accept rating.

**Limitations:**

yes

**Paper Formatting Concerns:**

NA.

**Quality:**

3

**Strengths And Weaknesses:**

**Strengths**
1. The idea of formulating the progressively unmasking image content in MGM as a multi-step decision problem is novel, and also aligns with the RL framework naturally.
2. In the state transition,  the proposed method treats newly unmasked tokens at each step as the decisive factor. Based on this, this paper proposes two new methods for calculating transition probabilities that are more suitable for MGM. The approaches seem effective from the experimental results.

**Weaknesses**
1. The paper claims a 38% performance improvement on GenEval over the baseline. The stated baseline appears to be Show-O 1.3B, but according to Table 2, the reported improvement is from 0.68 to 0.73. Could the 38% figure be incorrect?
2. In the ablation study, both of the proposed reduction strategies appear to significantly degrade performance, with results even falling below the Show-O baseline. Does this suggest that these two strategies may not be effective or meaningful?
3. Among the various evaluation metrics in GenEval, slight improvements are observed across all categories except for colors. However, the reward model used is CLIP, which is known to have limited capabilities in understanding object quantity and positional relationships, but a relatively strong ability to perceive color. How can this discrepancy be explained?
4. There are some typos that the author should address. For example, on line 79, "in-troduce" is incorrectly hyphenated. In the references on line 522, there's a formatting error. Figure 1 appears to be incomplete. In Figure 4, the horizontal axis is labeled "GenEval score," but the vertical axis represents the corresponding metrics; presenting this information in a table might be more effective.

---

> ### Author Rebuttal · Authors · 2025-07-30
>
> Dear Reviewer QEdg,
>
> We sincerely appreciate your thoughtful feedback and encouraging recognition of our novel idea and well-supported methodology. Your careful reading and insightful questions have been invaluable. Below we address each of your concerns in detail.
>
> # Q1: Table Statistics Error
> > The paper claims a 38% $\cdots$ be incorrect?
>
> Thank you for your careful review. The reported **38% improvement in GenEval performance is corret**, as shown in Figure 4. However, we sincerely apologize for an error in Table 2 of the original submission. Specifically, we **mistakenly reported an incorrect baseline score for Show-o**. The correct GenEval score of Show-o, as observed in our experiments, is 0.53, and our method achieves 0.73, resulting in the 38% improvement. Below is the corrected Table 2:
>
> **Table 2.** Model Comparison on Geneval. Results are from [1] or their original papers.
> | Model | Params | Single Obj. | Two Obj. | Counting | Colors | Position | Color Attri. | Overall |
> | ----- | ------ | ----------- | -------- | -------- | ------ | -------- | ------------ | ------- |
> | Show-o | 1.3B | 0.95 | 0.52 | 0.49 | 0.82 | 0.11 | 0.28 | 0.53 |
>
> Note that our experimental result is consistent with [1]. Similar variations are also found in other text-to-image (T2I) works[2,3]. We apologize for any confusion this may have caused and will revise Table 2 accordingly in the updated manuscript.
>
> # Q2: Reducing Strategy
> > In the ablation study $\cdots$ effective or meaningful?
>
> Thank you for raising this point. The GenEval scores for the two reduction strategies are:
>
> - Computation Reduction Strategy: 0.56
> - Unmasking Reduction Strategy: 0.66
>
> Both outperform the baseline (0.53), although they are below the Mask-GRPO (0.73). However, we emphasize that these strategies, especially the unmasking reduction strategy, are effective and meaningful due to **better sample effiency** and **lower training cost**. To be brief, **the unmasking reduction strategy achieves convergence using only 20% of the samples, resulting in a 25% performance improvement on GenEval while reducing training costs by 60%–80%.**
>
> We begin with a brief recap of the definition of our unmasking reduction strategy: it reduces the total number of unmasking iterations during training (in our experiment, from $T$ = 50 to $T$ = 10), while maintaining the full unmasking schedule during evaluation. Importantly, in GRPO-based training (our method), one sample does not correspond to a single prompt or image prompt or one image, but rather to a single unmasking iteration, since the loss is computed iteratively at each step (see Equation (10)). Therefore, by applying the unmasking reduction strategy, we reduce both the number of unmasking iterations and the sample count to just 20%. Despite using only 20% of the samples, this strategy delivers a 25% performance gain onGenEval. Moreover, it leads to a 60%~80% reduction in training cost. For these reasons, we conclude that the unmasking reduction strategy offers superior sample efficiency and is indeed effective.
>
> The computation reduction strategy also reduces training cost by computing loss over only  a subset of the total iterations. However, the unmasking conduction strategy is more effective than the computation reduction strategy, as detailed in Section 4.3.
>
> In conclusion, both strategies provide meaningful trade-offs between efficiency and performance, especially in online reinforcement learning (RL) settings。
>
> We appreciate your insightful question and hope these clarifications help.
>
> # Q3: Category Discrepancy
> > Among the various evaluation $\cdots$ discrepancy be explained?
>
> Thank you for pointing this out. First of all, we apologize again for the errors in Table 2. After correcting the baseline statistics (Q1), we confirm that our method improves GenEval performance across all categories, including colors. Please find the updated results below:
>
> **Table 2.** Model Comparison on Geneval. Results are from [1] or their original paper.
> |Model|Params|Single Obj.|Two Obj.|Counting|Colors|Position|Color Attri.|Overall|
> |-|-|-|-|-|-|-|-|-|
> |Show-o[2]|1.3B|0.95|0.52|0.49|0.82|0.11|0.28|0.53|
> |**Show-o + Mask-GRPO**|1.3B|**0.99**|**0.90**|**0.69**|**0.85**|**0.35**|**0.59**|**0.73**|
>
> However, we **acknowledge that the improvement in the 'Colors' category is relatively smaller**, while our reward model, clip model, has a strong ability to perceive color. We attribute this to the **limitation of the base model, which is the nature of RL-based post training**.
>
> Mask-GRPO is a RL-based post-training method in T2I generation. As discussed in recent works [4,5,6] on large language models, the power of RL-based post training is limited by the base model. RL is improving the performance by unlocking the potenial of the base model, 'teaching' it to 'climb higher' to where it is capable but does not know how to reach. However, RL cannot create new capabilities beyond the base model’s inherent capacity. Therefore, we believe the smaller gains in 'Colors' category reflect a performance ceiling of the base model in handling color semantics, which our method can optimize but not fundamentally surpass. We notice that [3] has similary Geneval results after applying RL to the same base model, which supports our explainations.
>
> Thank you again for your question. We hope we address the concern.
>
> # Q4: Typos and Figure 4
> > There are some typos $\cdots$ be more effective.
>
> Thank you for the suggestion. We will thoroughly review and correct all typos in the revision to ensure clarity and readability. We sincerely apologize for any inconvenience caused to your reading due to typos.
>
> Regarding Figure 4, we agree that presenting the results as a table improves clarity. We propose the following revised table:
>
> **Table 4.** Ablation performence on Geneval.
> | Method Varient | Geneval Score |
> | ----- | ------ |
> | Base Model | 0.53 |
> | + Mask-GRPO | 0.73 |
> | + Mask-GRPO w/KL | 0.62 |
> | + Mask-GRPO w/o Sample Filter | 0.60 |
> | + Mask-GRPO w/ Computation Reduction Strategy | 0.56 |
> | + Mask-GRPO w/ Unmasking Reduction Strategy | 0.66 |
>
> # References
> [1] Yan, Zhiyuan, et al. "Gpt-imgeval: A comprehensive benchmark for diagnosing gpt4o in image generation." arXiv preprint arXiv:2504.02782 (2025).
>
> [2] Guo, Ziyu, et al. "Can We Generate Images with CoT? Let's Verify and Reinforce Image Generation Step by Step." arXiv preprint arXiv:2501.13926 (2025).
>
> [3] Liu, Jie, et al. "Flow-grpo: Training flow matching models via online rl." arXiv preprint arXiv:2505.05470 (2025).
>
> [4] Gandhi, Kanishk, et al. "Cognitive behaviors that enable self-improving reasoners, or, four habits of highly effective stars." arXiv preprint arXiv:2503.01307 (2025).
>
> [5] Yue, Yang, et al. "Does reinforcement learning really incentivize reasoning capacity in llms beyond the base model?." arXiv preprint arXiv:2504.13837 (2025).
>
> [6] Shah, Darsh J., et al. "Rethinking reflection in pre-training." arXiv preprint arXiv:2504.04022 (2025).

---

> > ### Comment · Reviewer_QEdg · 2025-08-05
> >
> > I thank the authors for their efforts in the rebuttal. My questions have been addressed well. However, I also noticed that some typos were pointed out by other reviewers. I suggest the authors carefully review the manuscript to ensure it is free of any technical errors.

---

> ### Author Response · Authors · 2025-08-01
> **Clarification**
>
> Dear Reviewer QEdg,
>
> We would like to clarify a small error in our response to Q3. The reference in the sentence 'We notice that [3] has similary Geneval results after applying RL to the same base model' should be [2], not [3]. We sincerely apologize for any confusion this may have caused.
>
> Thank you once again for your time and thoughtful review. We look forward to hearing from you!

---

> ### Author Response · Authors · 2025-08-06
> **Typo Corrections**
>
> Dear Reviewer QEdg,
>
> Thank you for your valuable time and continued engagement with our work. We are pleased to hear that our previous response addressed your questions, and we sincerely appreciate you taking the extra step to provide suggestions on presentation. We have carefully addressed the point you raised. Please find the detailed revisions below.
>
> # Typos
>
> We sincerely apologize for the oversights in our initial submission and thank you for your meticulous review. We have **corrected all identified issues**. While the regulations prevent us from uploading an updated manuscript in Openreview, we assure you that all changes will be incorporated into the next version.
>
> - l.79
>
> Before: $\cdots$ first to in-troduce $\cdots$
>
> After: $\cdots$ first to introduce $\cdots$
>
> - l.196
>
> Before: $\cdots$  sampling the $k$-th at position $i$ at iteration $t$
>
> After: $\cdots$  sampling the $k$-th token at position $i$ at iteration $t$
>
> - l.229
>
> Before: $\cdots$ during training, e.g., from $T=50$ to $T=10$. while keeping $\cdots$
>
> After: $\cdots$ during training (e.g., from $T=50$ to $T=10$), while keeping $\cdots$
>
> - l.296
>
> Before: $\cdots$ are presneted in $\cdots$
>
> After: $\cdots$ are presented in $\cdots$
>
> - l.212
>
> Before: $\cdots$ effectively setting$\beta$=0 $\cdots$
>
> After: $\cdots$ effectively setting $\beta$=0 $\cdots$
>
> - l.235 : Added the figure for reward instability.
>
> - l.522 : Corrected reference formatting.
>
> - Figure 1 : Adjusted figure size and font size to match the main text.
>
> - Figure 4: Converted the figure content into a table, as detailed in our response to Q4.
>
> We are grateful for your attention to detail, which has helped us improve the quality and clarity of our manuscript.
>
> # Conclusion
>
> Once again, **thank you sincerely for your time, constructive feedback, and encouragement** throughout the review process. Your insightful suggestions have been invaluable in strengthening our manuscript. We hope our responses and revisions have fully addressed your concerns. If we have have successfully done so, **we would be very grateful if you would consider increasing your support in your final evaluation — Your support would be a tremendous encouragement to us**.
>
> Thank you!

---

> ### Author Response · Authors · 2025-08-08
> **Sincerely Looking Forward to Your Feedback**
>
> Dear Reviewer QEdg,
>
> We would like to sincerely thank you once again for your valuable feedback, insightful comments, and the time and effort you’ve dedicated to reviewing our work. Your constructive suggestions have been immensely helpful in improving our paper.
>
> As the discussion period is coming to a close in about 24 hours, we wanted to kindly check if our response has fully addressed your concerns. If there are any remaining questions or points for discussion, we would be more than happy to engage further.
>
> If our response has resolved your concerns, we would be truly grateful if you might consider increasing your support. Of course, we fully respect your decision either way and deeply appreciate your thoughtful review. Your support would mean a great deal to us!

---

> > ### Comment · Reviewer_QEdg · 2025-08-08
> >
> > Thank you for the efforts put into this work and the rebuttal. Considering the strengths and the overall discussion, I will maintain my positive rating and hope the authors will clarify the concerns raised by all reviewers in the revised version.

---

> ### Author Response · Authors · 2025-08-08
> **Thanks!**
>
> Dear Reviewer QEdg,
>
> Thank you once again for your response, your continued engagement, and your commitment to helping improve our paper. We truly appreciate the time and effort you have dedicated to reviewing our work! Until now it seems like we have addressed all concerns, and we will carefully clarify all of them in the revised version.

---

### Author Response · Authors · 2025-08-09
**Summary of the Rebuttal**

Dear Area Chair and Reviewers:

We would like to sincerely thank you once again for your time, effort, and valuable insights on our work. Your constructive comments and thoughtful suggestions have been extremely helpful. Below is a concise summary of our rebuttal and the key discussion points for your ease of reference.

# Reviewer Highlights from the Original Review

Our paper has been acknowledged for its clear motivation, strong novelty, solid methodology, comprehensive empirical evaluation, and well-organized presentation. Notable highlights include:

- Our idea of formulating the generation process in Masked Generative Models (MGMs) as a decision-making problem is **novel** (Reviewer QEdg, Vae1, 5UPe), with a **clear and natural motivation (All Reviewers)**

- Our method of applying Group Relative Policy Optimization (GRPO) to MGMs is
**applicable** (Reviewer QEdg), **well-grounded** (Reviewer 5UPe), **original**, and **well-motivated** (Reviewer Vae1, 5UPe). It is particularly interesting for **providing a principled way to bridge the training–inference mismatch** in MGMs (Reviewer Vae1) and **opens the door** for further RL applications in efficient, non-autoregressive generative frameworks. (Reviewer 5UPe)

- Our empirical results demonstrate the **superior effectiveness** of the proposed methods. (**All Reviewers**).

- Our presentation is generally **well-structured**. (Reviewer Ydx6, 5UPe)

# Addressing the Raised Concerns

We have **carefully reviewed and addressed all concerns**, including:

- Reward Model: Analysis of reward mechanisms (Q3 of Reviewer QEdg, Q2 of Reviewer Vae1), validation of reward model robustness (Q2 of Reviewer 5UPe).

- Transition Probability: More detailed explanations of transition probability definitions (Q1 of Reviewer Ydx6, Q1 of Reviewer 5UPe and furthur discussions)

- Proposed Enhancement Strategies: Deeper explanations of reduction strategies (Q2 of Reviewer QEdg, Q4 of Reviewer 5UPe), additional details on the sample filtering strategy (Q1 of Reviewer Vae1).

- Experiment: Clarifications on experimental results (Q1 of Reviewer QEdg), evaluation on additional benchmark (Q2 of Reviewer Ydx6).

- Base Model: Further evaluations on other MGMs (Q3 of Reviewer 5UPe).

- Other Aspects: Theoretical justification (Q5 of Reviewer 5UPe), prompts of generated images (Q3 of Reviewer Vae1), Extended limitations section (furthur discussion with Reviewer 5UPe), and corrections of typos.

Based on the reviewers' **positive feedback**, we believe that **we have thoroughly addressed all their concerns**.

# Reviewers’ Feedback about the Rebuttal

- Reviewer QEdg: "Thank you for the efforts put into this work and the rebuttal $\cdots$ **My questions have been addressed well**."

- Reviewer Vae1: "I find them **very insightful** $\cdots$ The CLIP model analysis is **particularly nice** $\cdots$ the rebuttal **gives food for thoughts** $\cdots$ I have no more comments and will re-assess."

- Reviewer Ydx6: "Thanks for your response. **I'll maintain my score. (positive)**"

- Reviewer 5UPe: "I appreciate the authors’ thorough and detailed rebuttal as well as the new experimental results $\cdots$ These additions **significantly strengthen the paper’s contributions and its claims of generalizability and robustness**."

Finally, we are truly grateful for the reviewers' constructive suggestions, which have significantly improved our work. We sincerely thank you once again for your time, careful review, and valuable feedback.

---

### Note · Authors · 2025-08-13

Dear Area Chair and Reviewers:

We would like to sincerely thank you once again for your time and effort in reviewing our work. We proposed Mask-GRPO, the first GRPO-based approach on MGMs for T2I generation, in this paper. Our paper received an **initial score of 4443**, and was recognized for its **clear motivation**, **strong novelty**, **solid methodology**, **comprehensive empirical evaluation**, and **well-organized presentation**. During the rebuttal period, we **carefully addressed all concerns** raised, and we have received **positive feedbacks from all reviewers**. These details are documented in our **previous general summary** submitted before the rebuttal deadline.

Below, we outline the **planned revisions** to further strengthen the paper:

# Additional Experiments

- Two more benchmarks: We evaluated our method on DPG-Bench and WISE, achieving 29% and 27% overall performance gains, respectively. These results demonstrate the generalizability of our approach beyond GenEval and MSCOCO-30K FID.

- Two more reward models: We conducted our method with other reward models, ImageReward and UnifiedReward, achieving 32% and 23% GenEval performance gains, respectively. These results demonstrate the adaptability and effectiveness of our approach across different reward models.

- Extension to another MGM: We have extended our method to another MGM - Meissonic, achieving up to 33% GenEval performance gain with various reward models. These results further demonstrate our method's generalizability across different MGM architectures.

- CLIP-based analysis: We conducted additional CLIP experiments to furthur reveal the underlying reward mechanisms.

# Detailed Explanations

- We will provide a more thorough explanation of transition probability definitions and reduction strategies.

- We will include more implementation details on the sample filtering strategy.

# Presentations

- We will expand the discussion of limitations in the 'Conclusion and Future Work' Section to better address the societal and ethical considerations, especially regarding reward models and reward hacking.

- We will correct all typos and formatting issues.

- We will replace the ambiguous term “newly predicted tokens” with clearer wording to avoid confusion.

We are deeply grateful for your constructive feedback, which has significantly improved the quality of our work. Thank you again for your time, careful review, and valuable suggestions.

---

### Decision · Program_Chairs · 2025-09-17

**Decision:**

Accept (poster)

**Comment:**

Summary of Reviews:

The paper introduces Mask-GRPO, a reinforcement learning (RL) method tailored for masked generative models (MGMs) in text-to-image (T2I) generation. By framing the unmasking process as a multi-step Markov decision problem, the method applies Group Relative Policy Optimization (GRPO) with novel transition probability definitions. Mask-GRPO incorporates enhancements like KL constraint removal, iteration reduction, and sample filtering, achieving improvements on benchmarks like GenEval and MSCOCO-30K FID.

Strengths of this paper include:
- Novelty: First RL approach designed specifically for masked generative models, bridging the training-inference mismatch in MGMs.
- Methodology: Reformulates MGM unmasking as a Markov decision process with principled transition probability definitions.
- Empirical Performance: Outperforms the baseline (Show-o) with significant improvements (e.g., 38% on GenEval, 10% on MSCOCO FID) while maintaining efficiency with a small model size (1.3B parameters).
- Practical Enhancements: Strategies like KL removal and sample filtering contribute to the model's robustness and scalability.

Weaknesses raised by the reviewers:
- Limited Theoretical Analysis: The paper lacks deeper theoretical justification for the stability and convergence of Mask-GRPO.
- Evaluation Scope: Experiments are confined to a single base model (Show-o) and benchmarks (GenEval, MSCOCO FID), with no exploration of newer datasets or benchmarks (e.g., DPG-Bench, WISE).
- Dependence on CLIP: Relies heavily on CLIP as the reward function without exploring alternative reward models, limiting generalizability.
- Technical Complexity: Equations (e.g., 14, 15) and some methodology details (e.g., transition probabilities) are difficult to follow and could benefit from clearer explanations.
- Manuscript Quality: Typographical errors, unclear figure labels, and insufficient details in some sections detract from the paper’s presentation.

Conclusion:
Mask-GRPO is a novel and promising approach for applying RL to MGMs in text-to-image generation, with strong empirical results and practical contributions. Concerns from reviewers mainly included the limited generalization, reliance on CLIP, and lack of theoretical analysis reduce its overall impact. During the rebuttal, reviewers are mostly satisfied by the added clarifications and experiments from the authors. Therefore, publication is recommended had the authors included these revisions and clarifications in the final version as they promised.